# CLIP THE BIAS: HOW USEFUL IS BALANCING DATA IN MULTIMODAL LEARNING?

**Ibrahim Alabdulmohsin[†], Xiao Wang[†], Andreas Steiner[†], Priya Goyal[△],**
**Alexander D'Amour[◇], Xiaohua Zhai[†]**
Google Deepmind: [†]Zürich, Switzerland. [△]New York, USA. [◇]Boston, USA.
`{ibomohsin,xzhai}@google.com`

## ABSTRACT

We study data-balancing for mitigating biases in contrastive language-image pre-training (CLIP), identifying areas of strength and limitation. First, we reaffirm prior conclusions that CLIP can inadvertently absorb stereotypes. To counter this, we present a novel algorithm, called Multi-Modal Moment Matching (M4), designed to reduce both representation and association biases in multimodal data. We use M4 to conduct an in-depth analysis taking into account various factors, such as the model, representation, and data size. Our study also explores the dynamic nature of how CLIP learns/unlearns biases. In particular, we find that fine-tuning is effective in countering representation biases, though its impact diminishes for association biases. Also, data balancing has a mixed impact on quality: it tends to improve classification but can hurt retrieval. Interestingly, data and architectural improvements seem to mitigate the negative impact of data balancing on performance; e.g. applying M4 to SigLIP-B/16 with data quality filters improves COCO image-to-text retrieval @5 from 86% (without data balancing) to 87% and ImageNet 0-shot classification from 77% to 77.5%! Finally, we conclude with recommendations for improving the efficacy of data balancing in multimodal systems.

## 1 INTRODUCTION

Recent advances on multimodal systems have been breathtaking, including in zero-shot classification (Radford et al., 2021; Zhai et al., 2022b; Yu et al., 2022a; Jia et al., 2021), text-to-image generation (Saharia et al., 2022; Yu et al., 2022b; Chang et al., 2023; Rombach et al., 2022; Ramesh et al., 2022), image captioning (Alayrac et al., 2022; Chen et al., 2022) and music generation (Agostinelli et al., 2023) to name a few. However, such systems can inflict harm if left unchecked, such as by amplifying biases, causing performance disparities, or encoding narrow cultural perspectives.

Contrary to the traditional supervised learning setup, which is well-understood, multimodal systems present novel ethical challenges. They typically operate with an *open-vocabulary*, in which the set of input and/or output tokens is unbounded. Hence, statistical definitions for bias such as demographic parity (Dwork et al., 2012; Zafar et al., 2017; Mehrabi et al., 2019) or equalized odds (Hardt et al., 2016; Kleinberg et al., 2016) do not extend easily to such systems and might even yield contradictory results under different setups (Akyürek et al., 2022). Second, *externalization* in multimodal systems – by allowing users to interact with the system in ways that can disclose its internal reasoning – along with the open-vocabulary nature can expose users to biases in unanticipated ways. Third, data in multimodal systems is potentially biased, perpetuating societal stereotypes (Birhane et al., 2021).

For concreteness, consider the following examples. If we query the text-to-image Imagen (Saharia et al., 2022) with the prompt: "*clipart picture of a manager dressed in black talking to a secretary dressed in blue,*" we get the generated image samples shown in Figure 1 (left), in which managers are men and secretaries are women. Similarly, if we ask for pictures of pilots and flight attendants, we get the sample of pictures shown in Figure 1 (right). Similar examples are reported by Wang et al. (2023), Cho et al. (2022), Tan et al. (2020) and Mishkin et al. (2022) using GANs (Goodfellow et al., 2014), Stable Diffusion (Rombach et al., 2022) and DALL-E (Ramesh et al., 2021).

Moreover, zero-shot classifiers are also susceptible to biases as highlighted by Hall et al. (2023) and Birhane et al. (2021), which we demonstrate in Figure 1 using contrastive image-language

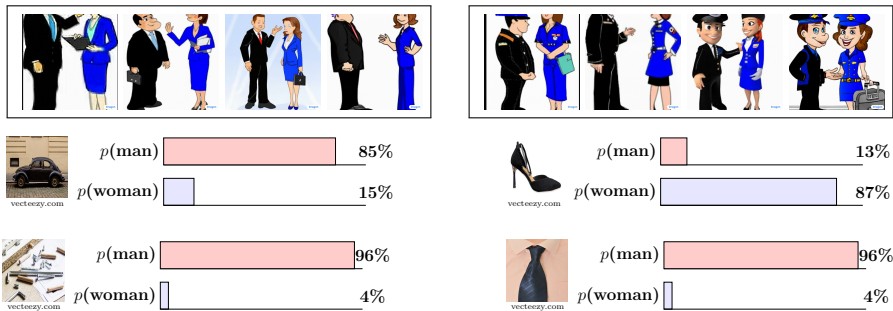

Figure 1: TOP: Text-to-image models prompted for occupations, such as manager / secretary (left) or pilot / flight attendant (right) can reflect societal stereotypes. Refer to Section 1 for the exact prompts. BOTTOM: CLIP can encode societal stereotypes, such as by associating cars with men. See Section 4.

pretraining (CLIP) (Radford et al., 2021). Evidently, CLIP encodes societal stereotypes, such as by associating ties, cars, and tools with men. Such unintended biases can inflict harm by shaping social narratives. In addition, due to their open vocabulary nature, users can apply such models in unanticipated ways, e.g. so-called "corner cases" (Birhane et al., 2021), to promote prejudices.

In this work, we do not claim to offer a comprehensive solution to the complex issues discussed above. Rather, we examine in depth the effectiveness of one remediation strategy: *data balancing*. Specifically, we investigate the impact of debiasing data in multimodal systems that align embeddings/representations across modalities in the contrastive learning setup.

We focus on contrastive learning for the following reasons. First, it captures many of the complexities involved in multimodal systems, including open vocabulary, externalization, lack of data distribution at inference time, and lack of well-established definitions of bias. Second, such systems exhibit strong societal biases (Hall et al., 2023; Birhane et al., 2021). Third, they are more amenable to analysis than generative models (e.g. when studied in the zero-shot visual classification setting or in cross-modal retrieval). Fourth, they are increasingly used in critical domains like healthcare (Sellergren et al., 2022; Tiu et al., 2022; Zhang et al., 2023), and are popular ingredients in many models, as seen in Florence (Yuan et al., 2021), CoCa (Yu et al., 2022a), and OWL-ViT (Minderer et al., 2022a). Finally, biases in such models can manifest downstream, e.g. in captioning and retrieval (Berg et al., 2022).

To enable this analysis, we develop a data balancing algorithm, called Multi-Modal Moment Matching (M4), and analyze it theoretically (see Section 5). Using M4, we conduct a comprehensive evaluation of the effectiveness of data balancing for obtaining desirable model behavior, identifying areas of strength and limitation, while accounting for factors such as the model, representation, and training data sizes. We also examine how quickly CLIP learns/unlearns biases, among other evaluations. In total, we train over 150 models. To the best of our knowledge, an empirical evaluation of this magnitude for data balancing in multimodal systems has never been conducted before (see Section 6). Broadly speaking, our results are nuanced, and suggest that while data balancing has a positive impact on the model bias with a mixed impact on its quality, it is insufficient for obtaining fair behavior downstream. This echoes prior observations in vision; e.g. Wang et al. (2019).

## 2 PRELIMINARIES

For a brief overview, CLIP contains two towers for vision and language, which are encoder-only transformers (Vaswani et al., 2017). Denoting $\mathbf{x} = (\mathbf{v}, \mathbf{t})$ for an image-text pair, let $(\mathbf{z}^v, \mathbf{z}^t) \in \mathbb{R}^r \times \mathbb{R}^r$ be the corresponding outputs of the two towers, each is embedded in $\mathbb{R}^r$. We refer to $r$ as the "representation size" in this paper. Given a fixed image $\mathbf{v}$ and a collection of captions $(\mathbf{t}_1, \ldots, \mathbf{t}_n)$ whose corresponding representations are $(\mathbf{z}_1^t, \ldots, \mathbf{z}_n^t) \in \mathbb{R}^{r \times n}$, CLIP assigns a probability score to each caption by first calculating the logits $l = (\langle \mathbf{z}^v, \mathbf{z}_k^t \rangle)_{k \in [n]} \in \mathbb{R}^n$ and then applying softmax normalization $p = \text{SoftMax}(l/T)$, where $T \in \mathbb{R}^+$ is a learnable temperature.

To study the effectiveness of data bias mitigation in CLIP, we introduce a data balancing algorithm that tackles biases in first-order statistics (such as perceived gender imbalances) and second-order statistics (such as correlating occupations with a particular perceived gender).

**Definition 1** (Data Representation Bias). *If* $\mathbf{s} \sim \mathcal{D} \in \{0, 1\}^m$ *is a sensitive attribute sampled i.i.d., the representation bias (RB) in* $\mathcal{D}$ *with respect to a target* $\pi \in [0, 1]^m$ *is:* $\max_{k \in [m]} |\pi_k - \mathbb{E}_{\mathcal{D}}[\mathbf{s}_k]|$.

To keep our analysis general, $\mathbf{s} \in \{0, 1\}^m$ in Definition 1 is a binary vector that encodes all (potentially overlapping) sensitive attribute categories, such as perceived gender and age groups. For instance, the first entry in $\mathbf{s}$ could be a binary indicator for the group "perceived women," the second a binary indicator for the group "perceived men," the next ten indicators for the Monk Skin Tone scale (Monk, 2019), and so on. We keep $\pi$ in our discussion arbitrary because the desired target distribution of categories may vary depending on the context, as discussed in (Berg et al., 2022).

Data representation bias (RB) measures differences in group prevalence; e.g. if $\pi = (0.5, 0.5)$ for "men" and "women," then having only "men" in 80% of the images implies a significant RB in the data. Definition 1 defines RB w.r.t. $\pi$ to account for overlapping attributes, such as gender and race.

**Definition 2** (Data Association Bias). *If* $(\mathbf{s}, \mathbf{y}) \in \mathcal{S} \times \mathcal{Y}$, *where* $\mathcal{S} = \{0, 1\}^m$ *and* $\mathcal{Y} = \{0, 1\}^c$, *is sampled i.i.d. from a joint distribution* $\mathcal{D}$, *association bias (AB) in* $\mathcal{D}$ *w.r.t.* $(\mathcal{S}, \mathcal{Y})$ *is defined as:*

$$\text{Data Association Bias} = \max_{k \in [m],\, r \in [c]} \left| \mathbb{E}_{\mathcal{D}} \left[ \mathbf{y}_r \,|\, \mathbf{s}_k = 1 \right] - \mathbb{E}_{\mathcal{D}} \left[ \mathbf{y}_r \,|\, \mathbf{s}_k = 0 \right] \right|. \tag{1}$$

Definition 2 is an extension of the widely-adopted notion of demographic parity (Dwork et al., 2012; Zafar et al., 2017; Alabdulmohsin & Lucic, 2021). It captures bias in second-order statistics. For example, if $\mathcal{Y}$ is the set of occupations and $\mathcal{S}$ is perceived gender, association bias is large when an occupation is more prevalent among "perceived men" compared to "perceived women."

Both types of bias in Definitions 1 and 2 are defined w.r.t. the data distribution. Next, we provide the analogous definitions for the model itself, which is always CLIP throughout our analysis.

**Definition 3** (Model Representation Bias). *If* $f : \mathcal{X} \to \Delta^m$ *is a classifier outputting a probability distribution over some sensitive attributes* $\mathcal{S}$, *the representation bias (RB) in* $f$ *w.r.t. a fixed target* $\pi \in [0, 1]^m$ *is:* $\max_{k \in [m]} |\pi_k - \mathbb{E}_{\mathcal{D}}[f_k(\mathbf{x})]|$, *where* $f_k$ *is the probability assigned to attribute* $k$.

In particular, if $\pi$ is uniform and the model assigns a higher probability to one group over others, on average, it has a larger representation bias. We care about RB because models should follow the *principle of indifference* (PI) (Eva, 2019) if images do not contain relevant evidence about subgroups.

**Definition 4** (Model Association Bias). *If* $(\mathbf{x}, \mathbf{s}, \mathbf{y}) \in \mathcal{X} \times \mathcal{S} \times \mathcal{Y} \sim \mathcal{D}$ *are drawn i.i.d., where the sensitive attribute* $\mathbf{s} \in \mathcal{S}$ *is as in Definition 2, the association bias (AB) in* $f : \mathcal{X} \to \mathcal{Y}$ *is:*

$$\text{Model Association Bias} = \max_{k \in [m],\, r \in [c]} \left| \mathbb{E}_{\mathcal{D}} \left[ f_r(\mathbf{x}) \,|\, \mathbf{s}_k = 1 \right] - \mathbb{E}_{\mathcal{D}} \left[ f_r(\mathbf{x}) \,|\, \mathbf{s}_k = 0 \right] \right|. \tag{2}$$

Our goal is to study how such biases in the data *transfer* to their corresponding biases in the model, and whether data balancing is impactful, especially in light of the *bias amplification* phenomenon often observed in the literature (Bolukbasi et al., 2016; Hendricks et al., 2018; Zhao et al., 2017). For consistency, we focus on perceived gender as a sensitive attribute $\mathbf{s}$ and use occupations as the set of labels $\mathcal{Y}$ in Definition 3. We denote the original data (without intervention) as `baseline`.

Generally, to mitigate biases in the data, we explore two options. The first option is to tackle the desired constraints *directly*; i.e. by focusing only on perceived gender and occupations. We refer to this version of the data as `balanced`. The second option is to include *proxies* as well. For instance, CLIP may associate "pilots" with "perceived men" even if decorrelated in the data because cockpits resemble machines, and "machines" might be associated with "perceived men" elsewhere in the data. The set of proxies we use, such as "computer" and "briefcase," is suggested by an LLM and listed in Appendix A.5. During training, we balance the marginal distribution of perceived gender and remove correlations with both occupation and proxies. We refer to this version of the data as `proxies`. In Appendix A.5.2, we motivate including proxies using the causal formalism in Veitch et al. (2021).

## 3  SUMMARY OF FINDINGS

Before presenting the detailed experimental results, we first summarize our major findings:

I **Proxies mitigate representation biases:** In addition to balancing the prevalence of sensitive attribute, decorrelating sensitive attributes against many proxy variables helps minimize the model's RB, thereby preventing the model from favoring certain subgroups in unrelated contexts.

II **But proxies hurt association biases:** For AB in the closed-vocabulary setting, e.g. removing correlation between gender and occupation, adding extra proxy variables can negatively affect attempts to reduce AB. This is likely because the added constraints compete with the original ones during data balancing when not all constraints can be satisfied simultaneously.

III **Fine-tuning is effective for representation bias:** RB in the model is sensitive to the *last* distribution seen on the training data. So, fine-tuning on balanced data effectively counters it.

IV **But fine-tuning is less effective for association bias:** The extent of AB in the model varies gradually based on the duration spent training on balanced data, irrespective of whether the balanced data is seen first or last during training.

V **Mixed impact on model performance:** Data balancing changes the distribution of human and non-human images/texts, which leads to improvement on visual classification benchmarks but worse performance on image-text retrieval. We explain this in Section 4.4 and Appendix A.6.

VI **But improving data quality and model architecture helps:** Improving data quality and model architecture, such as by filtering out image-text pairs with low similarity scores, seems to mitigate any potential negative impacts of data balancing on the model's performance. See Appendix A.8.

## 4 DETAILED RESULTS

**Setup.** We assume a potentially-infinite set of examples $(\mathbf{x}^{(i)}, \mathbf{y}^{(i)}, \mathbf{s}^{(i)})_{i \in \mathbb{N}}$, where $\mathbf{x}$ is an image-text pair, $\mathbf{y} \in \{0, 1\}^c$ are some predefined labels and $\mathbf{s} \in \{0, 1\}^m$ encodes sensitive attribute categories. See Section 5 for further details. We focus on perceived gender $\mathbf{s}$ and use occupations as the set of labels $\mathcal{Y}$ (see Definitions 1 and 2), both are inferred from image and text separately and then concatenated. For instance, to remove correlation between gender and occupation, we de-correlate all combinations of image- and text-based annotations. Since $\mathbf{s}$ and $\mathbf{y}$ are binary vectors, zero correlations imply independence so, by the data-processing inequality (Cover & Thomas, 1991), this offers a *stronger* bias guarantee than logical disjunction, where an attribute is marked present if detected in any modality. See Appendix A.1 for illustrations and additional discussions.

In order to explore the dynamic nature of how contrastive language-image pretraining (CLIP) learns and unlearns biases, we split training into two stages. In Stage 1, we train on either the original data without intervention or on the same dataset but after intervention. In Stage 2, we switch to the other version of the dataset. In total, each CLIP model is trained on 1B image-text pairs[1] from the WebLI dataset (Chen et al., 2022). We vary the length of Stage 1 in $\{0\%, 10\%, 90\%\}$. We use three random seeds for each setup. Appendix A.4 provides the full training configuration.

In order to study the architecture's impact, we experiment with two model sizes studied in Dosovitskiy et al. (2020) and Zhai et al. (2022a): (1) size **S** and size **B** with, respectively, 30M and 100M parameters for each modality. Besides, we also vary the representation size $r \in \{384, 768\}$ (see Section 2). Since we use a patch size of $32 \times 32$ in our sweep, we also verify key findings on larger sequence lengths using ViT-B/16 with $16 \times 16$ patch sizes. We conduct a meta-analysis on all configurations to determine statistical significance utilizing the Wilcoxon's signed-rank test (Wilcoxon, 1992).

### 4.1 REPRESENTATION BIAS

To evaluate RB in CLIP according to Definition 3, we use ILSRCV2012 (Deng et al., 2009) and compare the parity $\mathbb{E}[p(\text{``man''}) - p(\text{``woman''})]$ across a random subset of 8K images. Intuitively, models should be indifferent to perceived gender in the absence of evidence, as discussed in Section 2.

**Single Distribution.** Figure 2 (top) summarizes the results for models trained on a single distribution. First, we observe that the amount of training examples seems to offer little benefit; mean parity cannot be reduced by simply training on bigger datasets. Second, balancing the data – with or without proxies – helps in mitigating RB. In fact, adding proxies seems particularly beneficial for models with large representations trained on large datasets. Refer to Appendix A.7 for additional figures.

---

[1]Because debiasing removes about $10\%$ of the examples, models trained on debiased data see some examples twice. However, there is evidence that examples seen twice behave like fresh examples when training is not converged (Alabdulmohsin et al., 2022b) so we do not expect this difference to have an impact on the results.

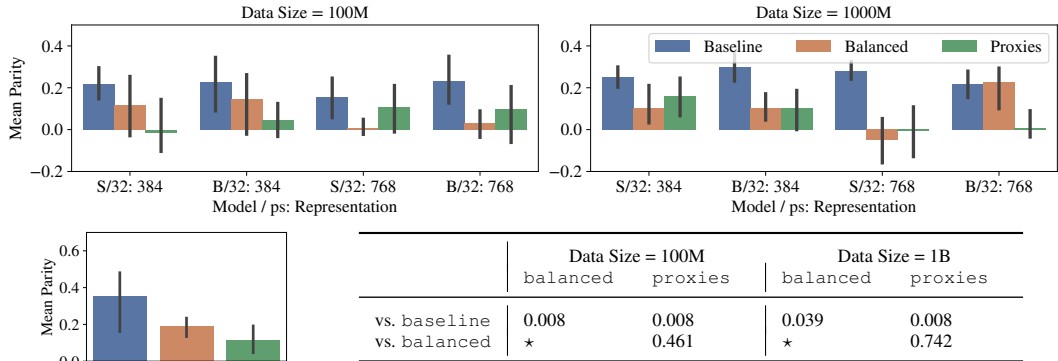

Figure 2: TOP: Mean parity $\mathbb{E}[p(\mathrm{man}) - p(\mathrm{woman})]$ across images from the ILSRCV2012 dataset (Deng et al., 2009). Values closer to zero are better. BOTTOM: On left, parity scores for ViT-B/16 (longer visual sequence length). On right, $p$ values calculated using Wilxocon's signed rank test (Wilcoxon, 1992) for the null hypothesis that column has the same effect as row.

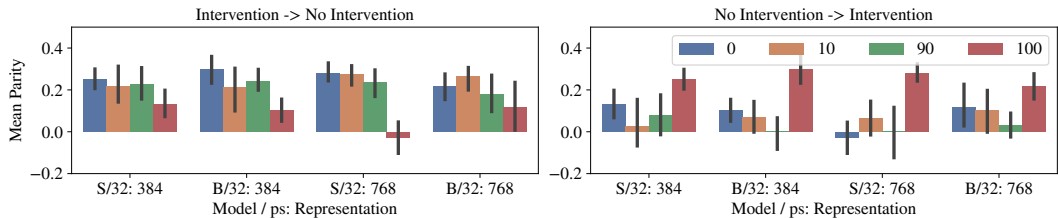

Figure 3: CLIP is trained on 1B examples split into two stages. On the left, it is initially trained on intervened data with proxies, before switching to the original data. On the right, it is trained on the original data before intervening. Legends indicate the fraction of time [%] assigned to Stage 1.

**Learning Dynamics.** Figure 3 displays the mean parity but for different durations of Stage 1. To recall, we split training into two stages in order to analyze how quickly CLIP learns and unlearns biases. As shown in Figure 3, the *last* distribution seen by the model, even if it is a meager 10% of the training duration, heavily impacts its parity. So, fine-tuning on balanced data is an effective remedy.

## 4.2 ASSOCIATION BIAS

Our second set of evaluations examine the association bias (AB) between perceived gender and occupations. We use FairFace (Karkkainen & Joo, 2021), UTKFace (Zhang et al., 2017) and MIAP (Schumann et al., 2021) datasets, all annotated with perceived gender attributes. The first two datasets contain face images while MIAP contains images more representative of real-world scenarios. We include MIAP because cultural biases extend beyond faces to artifacts and social practices (Berg et al., 2022). To evaluate AB, we calculate the mean absolute parity $(1/|\mathcal{Y}|) \sum_{\mathbf{y} \in \mathcal{Y}} |p(\mathbf{y}|\mathrm{man}) - p(\mathbf{y}|\mathrm{woman})|$ across all occupations $\mathcal{Y}$ by providing CLIP with two labels: the occupation's name vs. the empty string. Then, we average those for all images of perceived men and all images of perceived women. In MIAP, only images containing a single perceived gender are used.

**Single Distribution.** Figure 4 (top) summarizes the average parity in models trained on a single data distribution. First, training longer increases the level of bias in `baseline`, likely because longer training allows the model to reflect biases in the data. But, we again observe that balancing data helps. Nevertheless, adding proxies seems to hurt, likely because the added constraints compete when not all constraints during data balancing can be satisfied simultaneously.

**Learning Dynamics.** Figure 5 plots AB vs. the time of intervention. Unlike in RB (see Section 4.1), we now observe a gradual increase or decline in AB depending on how long the model is trained on intervened data, irrespective of whether the balanced data is seen first or last during training.

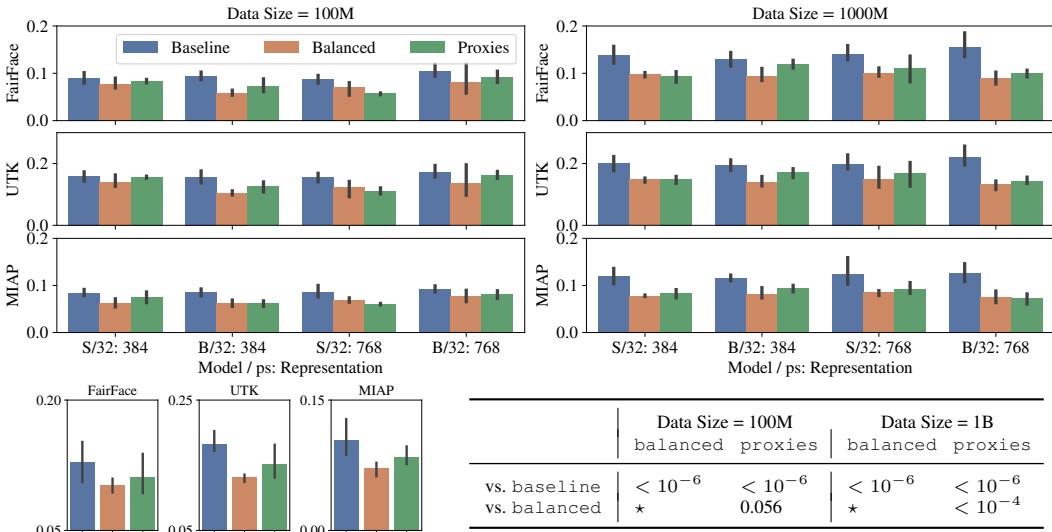

Figure 4: TOP: A comparison of AB (perceived gender against occupation) evaluated in three downstream datasets. BOTTOM: ViT-B/16 results (left) and statistical analysis (right) as in Figure 2.

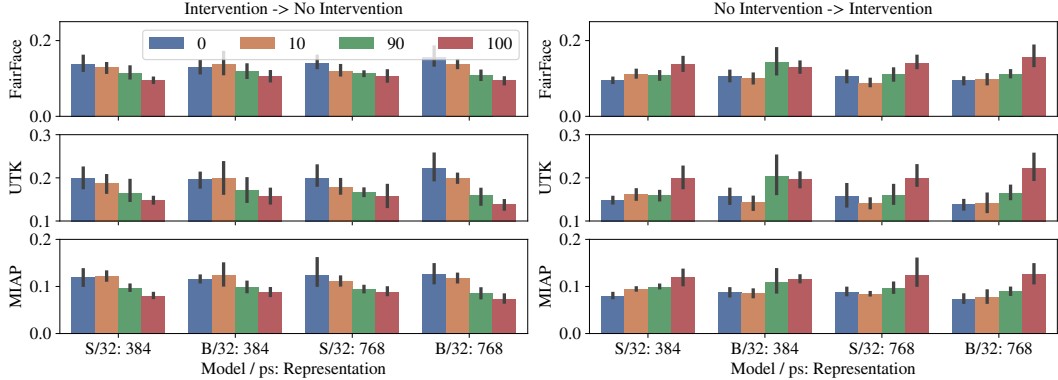

Figure 5: A summary of how CLIP learns or unlearns association bias (shown in $y$-axis) when intervened data comprises different percentages [%] of training duration. Setup is similar to Figure 3.

### 4.3 RECOGNIZING SENSITIVE ATTRIBUTES

The simplest way for a model to remove its bias is to be entirely blind or unaware of the sensitive attribute. For contrastive models, such outcome is undesirable since it impacts utility; e.g. the model may not distinguish between "father" or "mother" in the caption. To examine if the improvement in RB and AB due to data balancing impacts the ability of the model to recognize those attributes, we use the zero-shot classification setting with the two labels "man" and "woman" for each image in FairFace, UTKFace and MIAP. When aggregating errors using Wilcoxon's signed rank test (Wilcoxon, 1992) with the null hypothesis that (baseline, balanced, proxies) yield the same performance, we have $p > 0.05$, indicating the absence of statistically significant differences. See Appendix A.7.

### 4.4 MODEL QUALITY

Next, we compare the quality of CLIP models across three downstream evaluations: zero-shot classification, few-shot image classification using the representation provided by the image tower, and retrieval for both COCO (Lin et al., 2015) and Flickr (Young et al., 2014). Here, we report results for ViT-B/16 image tower and defer the full set of figures to Appendix A.7. As shown in Appendix A.7, Table 1 and Figure 6, balancing the data improves classification, on average, but

Table 1: 10-shot classification results using ViT-B/16 as image tower in CLIP, pretrained on 1B examples. Datasets are ILSRCV2012 (Deng et al., 2009), Colorectal (Kather et al., 2016), Cars (Krause et al., 2013), Birds (Welinder et al., 2010), UC (Yang & Newsam, 2010), CIFAR100 (Krizhevsky, 2009), DTD (Cimpoi et al., 2014), Caltech (Fei-Fei et al., 2004) and Pets (Parkhi et al., 2012).

|          | INet | Color | Cars | Birds | UC | C100 | DTD | Caltech | Pets |
|----------|------|-------|------|-------|-----|------|-----|---------|------|
| baseline | $50.4^{\pm.1}$ | $\mathbf{75.7}^{\pm.8}$ | $77.5^{\pm.3}$ | $46.8^{\pm.6}$ | $91.3^{\pm.1}$ | $\mathbf{57.8}^{\pm.6}$ | $66.9^{\pm.1}$ | $88.9^{\pm.1}$ | $71.9^{\pm.1}$ |
| balanced | $50.9^{\pm.1}$ | $74.9^{\pm.6}$ | $\mathbf{79.0}^{\pm.1}$ | $47.6^{\pm.2}$ | $\mathbf{92.2}^{\pm.3}$ | $57.4^{\pm.9}$ | $\mathbf{67.4}^{\pm.3}$ | $89.2^{\pm.3}$ | $73.3^{\pm.4}$ |
| proxies  | $\mathbf{51.1}^{\pm.1}$ | $75.0^{\pm.6}$ | $78.7^{\pm.2}$ | $\mathbf{47.7}^{\pm.1}$ | $91.5^{\pm.3}$ | $57.2^{\pm.2}$ | $67.0^{\pm.2}$ | $\mathbf{89.6}^{\pm.1}$ | $\mathbf{73.4}^{\pm.2}$ |

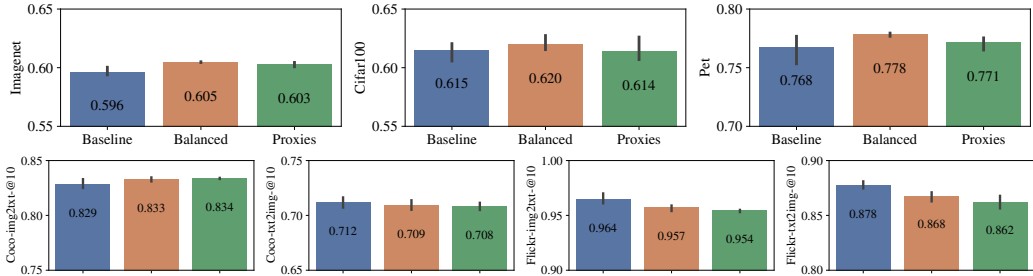

Figure 6: TOP: Zero-shot classification for ILSRCV2012, CIFAR100, and Pets. BOTTOM: Retrieval results (image-to-text and text-to-image) for COCO and Flickr. See Appendix A.7 for full results.

hurts retrieval. In addition, the impact on each metric is statistically significant with $p < 10^{-5}$. In Appendix A.6, we conduct an in-depth analysis, which reveals that the impact on quality is attributed to the distribution shift of human and non-human image-text pairs. In particular, debiasing reduces the number of examples containing humans, which improves visual classification since most benchmarks like ImageNet contain few (if any) human images. By contrast, retrieval datasets, such as COCO, contain a significant number ($> 40\%$). We analyze and reproduce the effect of debiasing in A.6.

**Data Quality & Architectural Improvement.** In Appendix A.8, we show that improving the data quality and architecture, by grounding texts on images and using the recent Sigmoid-loss Language-Image Pretraining (SigLIP) (Zhai et al., 2023), helps in mitigating any potential negative impacts of data balancing on the model's performance.

## 5 MULTI-MODAL MOMENT MATCHING (M4)

Denote $\mathcal{S} : |\mathcal{S}| = m$ and $\mathcal{Y} : |\mathcal{Y}| = c$ for the set of attributes and labels respectively. To mitigate representation and association biases (see Section 2), we reweigh examples in the data. It can be shown that finding a set of weights $\mathbf{q}$ assigned to training examples, which mitigate the two types of bias, is equivalent to finding a feasible solution $\mathbf{q}$ to the following constraints (see Appendix A.2.1):

$$\forall_{k\in[m],\,r\in[c]} \left| \mathbb{E}_{\mathcal{D}} \left[ \mathbf{q} \cdot (\mathbf{s}_k - \pi_k) \cdot \mathbf{y}_r \right] \right| \le \epsilon_D \quad \wedge \quad \forall_{k\in[m]} \left| \mathbb{E}_{\mathcal{D}} \left[ \mathbf{q} \cdot (\mathbf{s}_k - \pi_k) \right] \right| \le \epsilon_R, \quad (3)$$

for some small tolerance levels $\epsilon_D$ and $\epsilon_R$. The intuition behind this follows from the fact that since both $\mathbf{y}_r$ and $\mathbf{s}_k$ are binary-valued, zero covariance implies independence. Because finding a solution when $\mathbf{q} \in \{0, 1\}$ is NP-hard (Mehrotra & Celis, 2021), we relax the problem by optimizing $\mathbf{q}$ within the unit interval $[0, 1]$ and we sub-sample from the data according to it. However, since the same algorithm can potentially be used in-processing, with $\mathbf{q}$ taking any value in $\mathbb{R}^+$, we make our treatment general by adding the constraints $0 \le \mathbf{q} \le Q$ for some $Q \in \mathbb{R}^+ \cup \{\infty\}$.

In practice, it can sometimes be useful to constrain the *average* size of $\mathbf{q}$ as well. For example, when subsampling from a dataset, $\mathbb{E}[\mathbf{q}]$ must be equal to the subsampling rate $\eta \in (0, 1)$. In addition, a common assumption in fair subset selection is to allow examples to have different utilities $\mathbf{u} \ge 0$ (Stoyanovich et al., 2018); e.g. based on video engagement or text/image quality. Finally, bias constraints should correspond to a *soft* penalty to accommodate cases when they are not feasible, which can occur even when $\epsilon_D, \epsilon_R > 0$. We incorporate all such features into the algorithm.

| **Algorithm 1:** Update step per example | **Algorithm 2:** Bias vector implementation. |
|---|---|
| **Hyperparameters**: | |

**Algorithm 1:** Update step per example

**Hyperparameters**:

| | | |
|---|---|---|
| $\eta \in \mathbb{R}^{++}$ | : | average weight size |
| $Q > \eta \in \mathbb{R}^{++}$ | : | maximum weight |
| $V \in \mathbb{R}^{++}$ | : | enforcement level |
| $\tau \in \mathbb{R}^{++}$ | : | learning rate |
| $\epsilon_D, \epsilon_R \in [0, 1)$ | : | bias constraint levels |

**Update**:
1: $\mathbf{a} \leftarrow \text{biasVector}(\mathbf{s}, \mathbf{y}, \pi, \epsilon_D, \epsilon_R)$
2: $\beta \leftarrow [v^T \mathbf{a} + \mu - \eta \mathbf{u}]^+$
3: $\alpha \leftarrow [\mathbf{u}(\eta - Q) - v^T \mathbf{a} - \mu]^+$
4: $\mathbf{q} \leftarrow \eta - (v^T \mathbf{a} + \mu + \alpha - \beta)/\mathbf{u}$
5: $v \leftarrow v + \tau \frac{\mathbf{q}}{\eta} \mathbf{a}$
6: $\mu \leftarrow \mu + \tau \left( \frac{\mathbf{q}}{\eta} - 1 \right)$

**Algorithm 2:** Bias vector implementation.

```
def biasVector(s, z, pi, epsd, epsr):
    """
    Args:
     s: sensitive attributes, of length m.
     z: labels, of length c.
     pi: target distribution, of length m.
     epsd & epsr: bias constraint levels.

    Returns:
     b: bias vector of length 2m(c+1).
    """
    dp = np.einsum("bi,bo->bio", s - pi, z).
        reshape((batch_size, -1))
    b = np.concatenate([
        dp - epsd, -dp - epsd, # AB
        s - pi - epsr, -s + pi -epsr # RB
    ], axis=1)
    return b
```

Figure 7: Pseudo-code of the data balancing algorithm in Section 5. LEFT: Single update per example $(\mathbf{s}, \mathbf{y}, \mathbf{u})$, where $\mathbf{u}$ is the example's utility. RIGHT: Numpy-like implementation of the bias vector $\mathbf{a}$.

An overview of the data balancing algorithm is shown in Figure 7. It maintains two optimization variables $v \in \mathbb{R}^{2m(c+1)}$ and $\mu \in \mathbb{R}$, which are used to calculate the sample weight $\mathbf{q}$ by solving:

$$\underset{\mathbb{E}[\mathbf{q}]=\eta \,\wedge\, 0 \leq \mathbf{q} \leq Q}{\text{minimize}} \left\{ \frac{1}{2} \mathbb{E}_{\mathcal{D}} \left[ \mathbf{u} \cdot (\mathbf{q} - \eta)^2 \right] + V \cdot \left( \sum_{k \in [m]} l_k^R + \sum_{k \in [m],\, r \in [c]} l_{k,r}^D \right) \right\}, \quad (4)$$

where $l_{k,r}^D = \max \left\{ 0, |\mathbb{E}_{\mathcal{D}} [\mathbf{q} \cdot (\mathbf{s}_k - \pi_k) \cdot \mathbf{y}_r]| - \epsilon_{DP} \right\}$ and $l_k^R = \max\{0, |\mathbb{E}_{\mathcal{D}} [\mathbf{q} \cdot (\mathbf{s}_k - \pi_k)]| - \epsilon_R\}$ are the violations to the bias constraints in (3). The first term in (4) encourages the weights to be close to $\eta$ since we have the constraint $\mathbb{E}[q] = \eta$ while assigning higher priority to examples with greater utility $\mathbf{u}$. The second term penalizes biases with $V > 0$ controlling bias enforcement levels.

**Proposition 1.** *Algorithm 1 terminates with an optimal solution to the optimization problem in (4).*

The proof is in Appendix A.2.1. At inference time, the weight $\mathbf{q}$ assigned to a new example is:

$$\mathbf{q} = \eta - \frac{1}{\mathbf{u}} \left( v^T \mathbf{a} + \mu + \left[ \mathbf{u} (\eta - Q) - v^T \mathbf{a} - \mu \right]^+ - \left[ v^T \mathbf{a} + \mu - \eta \mathbf{u} \right]^+ \right). \quad (5)$$

Here, $\mathbf{a}$ is a "bias vector" calculated from the sensitive attributes $\mathbf{s}$ and labels $\mathbf{y}$ (see Appendix A.2.1). We provide examples of how the algorithm works and empirical verification in Appendix A.2.3. We also compare Algorithm 1 against other debiasing methods for binary classification in Appendix A.2.4.

**Proposition 2.** *Starting from the initial values $v = 0$ and $u = 0$, let $F_t$ be the dual loss of the optimization problem in (4) after $t$ updates of Algorithm 1 and denote $F_\infty$ for its limiting value. Then, Algorithm 1 with the learning rate schedule $\tau_t = O(1/\sqrt{t})$ satisfies:* $\left| \min_t \mathbb{E}[F_t] - F_\infty \right| \leq O\left( \left( \frac{Qmc}{\eta} \right)^2 \frac{\log t}{\sqrt{t}} \right).$

The proof is in Appendix A.2.2. Proposition 2 states that Algorithm 1 needs, at most, $O(Qmc/\eta)^4$ examples to converge. Since $m = |\mathcal{S}|$ and $c = |\mathcal{Y}|$ are typically small, convergence is fast. Throughout our experiments, we use the weights $\mathbf{q}$ to *subsample* from the original dataset. The subsampling rate ($\eta$ in Algorithm 1) is chosen to be the maximum rate where bias constraints are still satisfiable, which happens to be 90% in our setup. We use subsampling, instead of reweighting, because subsampling tends to perform more favorably (Sagawa et al., 2020; Celis et al., 2018).

## 6 RELATED WORKS

**Fairness in Multimodal Systems.** Fairness is a social construct and an important consideration when evaluating machine learning models. Research has shown that in the absence of bias mitigation, machine learning systems can amplify societal stereotypes (Hendricks et al., 2018; Bolukbasi et al.,

2016; Caliskan et al., 2017; Yang et al., 2020), cause performance disparities (Buolamwini & Gebru, 2018; Deuschel et al., 2020) and encode cultural biases and perspectives (Hutchinson et al., 2022; DeVries et al., 2019). For multimodal systems, in particular, while there is a growing interest in their applications and datasets, such as LAION (Schuhmann et al., 2021) and WebLI (Chen et al., 2022), recent findings indicate that the use of multiple modalities not only continues to encode societal biases (Hutchinson et al., 2022; Wang et al., 2023) including in CLIP models (Hall et al., 2023; Wang et al., 2022), but can also *amplify them further* compared to unimodal systems (Booth et al., 2021). This includes not only text-to-image generative models and CLIP but also image captioning (Zhao et al., 2021; Tang et al., 2021). In particular, multimodal datasets, such as LAION, were found to contain problematic content, such as stereotypes and ethnic slurs (Birhane et al., 2021). Few methods have been proposed for mitigating such biases including adversarial training (Yan et al., 2020; Berg et al., 2022), projection (Wang et al., 2023) and dropout (Wang et al., 2021). Yet, we are not aware of prior works that examine the effectiveness of data balancing, such as in CLIP (Radford et al., 2021).

**Reweighting methods.** The data balancing algorithm we develop is a variant of reweighting methods. Because it does not alter examples (unlike fair representation learning such as Zemel et al. (2013); Feldman et al. (2015); Lum & Johndrow (2016); Calmon et al. (2017); Madras et al. (2018)) and does not alter labels (unlike methods such as Kamiran & Calders (2009) and Alabdulmohsin et al. (2022a)), it serves as a viable baseline for debiasing CLIP-style models. However, there has been some skepticism in parts of the literature about the efficacy of reweighting in overparameterized models (Byrd & Lipton, 2019). Intuitively, an overparameterized model has the potential to perfectly fit all training examples, rendering it optimal regardless of the sample weights used. Nonetheless, in the multimodal setting, the size of the training dataset can be extremely large (i.e., billions of examples), and the model is only trained for a few epochs. In this setup, re-weighting the data may can be impactful as we demonstrate in our experiments. In the traditional supervised learning setup, there is evidence that reweighting is competitive (Chen et al., 2018; Idrissi et al., 2022; Choi et al., 2020), and that subsampling performs more favorably (Sagawa et al., 2020; Celis et al., 2018). Analogous techniques have additionally been used in other contexts, including texts (Dixon et al., 2018) and fine-tuning on balanced data to mitigate spurious correlations (Kirichenko et al., 2022).

**Advantages of M4.** Our data diversity algorithm offers more flexibility than prior methods by relaxing many of their constraints. For instance, it accommodates an arbitrary number of *overlapping* groups and attributes, making it also capable of handling traditional multiclass settings for which few black-box debiasing algorithms exist and they have only recently been developed (Alghamdi et al., 2022; Alabdulmohsin et al., 2022a). Diversity sampling has a well-established history, sometimes referred to as "fair subset selection" (Drosou et al., 2017; Mehrotra & Celis, 2021). It originates from early observations in search that sub-sampling data by maximizing utility leads to an under-representation of minorities (Kay et al., 2015). However, prior works, such as Stoyanovich et al. (2018), Yan et al. (2020) and Mehrotra & Celis (2021), only address representational biases (i.e., first-order statistics) and fail to account for association biases (i.e., second-order statistics). Other data balancing algorithms such as in Mehrotra & Celis (2021) even solve LPs, making them prohibitive for Internet-scale multimodal data. The data balancing algorithm we introduce resolves such limitations.

# 7 CONCLUSION AND FUTURE WORK

We present a data balancing algorithm and use it to study the impact of mitigating biases in Contrastive Language Image Pretraining (CLIP), a popular training paradigm used in several domains. We define representation and association bias in data and model separately and carefully disentangle the effects of the model, data, and representation size. Our findings suggest that balancing data is impactful but insufficient for obtaining fair downstream behavior so it should be combined with other intervention methods, such as in- and post-processing. In addition, it is recommended that models are trained on balanced data from the outset given that fine-tuning is less effective in removing association biases. Finally, the impact on quality should be assessed for both human-related metrics (e.g. describing actions) and non-human related metrics (e.g. classifying objects) since they can behave differently. Nevertheless, our analysis is necessarily limited since fairness cannot be reduced to statistical metrics. For example, we do not discuss which sensitive attributes are relevant or what acceptable label associations might be, and we only focus on contrastive (not generative) models. Potential future directions include exploring data augmentation methods and validating the effectiveness of our work on datasets such as Casual Conversations (Hazirbas et al., 2021), which have self-identified labels.

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

Table 2: Dataset examples. The first two columns contain the image (resized to 224px) and text (translated to English), as seen by the model. The third and fourth column show the *profession* and *gender* attributes that are extracted from the image (via an image classifier) and text (via regular expression matching), respectively.

| Image | Text | $\mathbf{s}^v, \mathbf{y}^v$ | $\mathbf{s}^t, \mathbf{y}^t$ | Comments |
|---|---|---|---|---|
|  pexels.com | female judge sentencing | | judge perceived woman | Example of an image where the image signals are incomplete, and the text signals provide additional information that would otherwise be lost. |
|  pixabay.com | archery bow and arrow aim | sport perceived woman | | The gender is not mentioned in the text description, and the regular expression does not trigger for "sport", but the image signal extracts both sensitive attribute and profession. |
|  vecteezy.com | a male doctor wearing a green scrub sits on a couch and is falling asleep | perceived woman | doctor perceived man | Here the text introduces an additional association between the profession "doctor" and the gender, which was not extracted from the image alone. |
|  pexels.com | photo of man surfing during daytime | sport | perceived man | In this case, the image signal only points towards a profession, but the text adds the association with "perceived men". |

## A  APPENDIX

### A.1  SENSITIVE ATTRIBUTE ANNOTATION

We infer attributes from both modalities. Using a vision classifier for images and regular expression matching on captions. Image modality is needed because even highly reputable websites have poor alt-text coverage (Bigham et al., 2006). Using the text modality is needed because images could be interpreted incorrectly noisy annotators. For example, (Birhane et al., 2021) shows how a photograph of a female astronaut could be misinterpreted by a image classifier as a picture of a "housewife in an orange jumpsuit." Hence, both modalities are used in our analysis.

**Why do we infer attributes from both modalities?**  As demonstrated in Table 2, the extracted signals from image and text can be incomplete, and in some cases one of the modalities creates an association that is not present in the image or text alone.

**Why do we concatenate them?**    As explained in Section 4, we *concatenate* annotations from both modalities. For instance, to remove correlation between gender and occupation, we de-correlate all combinations of image- and text-based annotations. Since $\mathbf{s}$ and $\mathbf{y}$ are binary vectors, zero correlations imply independence so, by the data processing inequality (Cover & Thomas, 1991), this offers a stronger bias guarantee than logical disjunction, where an attribute is marked present if detected in any modality. We illustrate this with a synthetic example next.

Consider the example shown in the table below, where we have 8 examples and the annotations of both $\mathbf{s}$ and $\mathbf{y}$ are provided from both image and text modalities.

|       | $\mathbf{s}$ (image) | $\mathbf{y}$ (image) | $\mathbf{s}$ (text) | $\mathbf{y}$ (text) | $\mathbf{s}$ (image or text) | $\mathbf{y}$ (image or text) |
|-------|------|------|------|------|------|------|
| Ex 1  | 1 | 1 | 1 | 0 | 1 | 1 |
| Ex 2  | 1 | 0 | 1 | 0 | 1 | 0 |
| Ex 3  | 1 | 1 | 0 | 1 | 1 | 1 |
| Ex 4  | 1 | 0 | 1 | 0 | 1 | 0 |
| Ex 5  | 0 | 0 | 0 | 1 | 0 | 1 |
| Ex 6  | 0 | 0 | 0 | 0 | 0 | 0 |
| Ex 7  | 0 | 0 | 0 | 1 | 0 | 1 |
| Ex 8  | 0 | 0 | 0 | 0 | 0 | 0 |

In this example, if we assume that an attribute is present when it is detected by any modality, we do not observe any bias in the data (rightmost two columns). In particular, the correlation between $\mathbf{s}$ and $\mathbf{y}$ is zero and both attributes are statistically independent of each other. However, both the image and text towers see different pictures, where $\mathbf{s}$ and $\mathbf{y}$ are correlated with an absolute correlation coefficient exceeding $|\rho| > 0.5$. Hence, by ensuring that $\mathbf{s}$ (image) is decorrelated from both $\mathbf{y}$ (image) and $\mathbf{y}$ (text) and vice versa, a stronger bias guarantee is ensured.

## A.2 DATA BALANCING ALGORITHM

### A.2.1 PROOF OF PROPOSITION 1

First, we prove that finding a sample weight function $\mathbf{q}$ that mitigates both types of bias in Definitions 1 and 2 is equivalent to finding a solution to the set of constraints in (3).

Since $\mathbf{y}_r$ and $\mathbf{s}_k$ are both binary-valued random variables, having zero demographic parity (DP) is equivalent to zero covariance (Alabdulmohsin & Lucic, 2021) so the DP condition is equivalent to:

$$\mathbb{E}\left[\mathbf{q} \cdot (\mathbf{s}_k - \pi_k) \cdot \mathbf{y}_r\right] = 0 \tag{6}$$

because:

$$\mathbb{E}\left[\mathbf{q} \cdot (\mathbf{s}_k - \pi_k) \cdot \mathbf{y}_r\right] = \mathbb{E}\left[\mathbf{q} \cdot \mathbf{s}_k \cdot \mathbf{y}_r\right] - \pi_k \mathbb{E}\left[\mathbf{q} \cdot \mathbf{y}\right] = \mathbb{E}\left[\mathbf{q} \cdot \mathbf{s}_k \cdot \mathbf{y}_r\right] - \frac{\mathbb{E}[\mathbf{q} \cdot \mathbf{s}_k]}{\mathbb{E}[\mathbf{q}]} \cdot \mathbb{E}\left[\mathbf{q} \cdot \mathbf{y}\right]$$

By dividing both sides by $\mathbb{E}[\mathbf{q}]$, we have:

$$\frac{\mathbb{E}\left[\mathbf{q} \cdot (\mathbf{s}_k - \pi_k) \cdot \mathbf{y}_r\right]}{\mathbb{E}[\mathbf{q}]} = \frac{\mathbb{E}\left[\mathbf{q} \cdot \mathbf{s}_k \cdot \mathbf{y}_r\right]}{\mathbb{E}[\mathbf{q}]} - \frac{\mathbb{E}[\mathbf{q} \cdot \mathbf{s}_k]}{\mathbb{E}[\mathbf{q}]} \cdot \frac{\mathbb{E}\left[\mathbf{q} \cdot \mathbf{y}\right]}{\mathbb{E}[\mathbf{q}]},$$

which is the covariance w.r.t. the sample weights $\mathbf{q}$. Hence, if condition (6) is satisfied, the association bias is zero.

Second, satisfying the representation bias (RB) constraints is equivalent to setting $\pi_k = \mathbb{E}[\mathbf{q} \cdot \mathbf{s}_k]/\mathbb{E}[\mathbf{q}]$ by definition, which is equivalent to the second condition in (3).

To solve the optimization problem (4), we solve the dual problem. To simplify notation, suppose initially that the dataset is finite and let $q \in \mathbb{R}^n$ be a vector of length $n$ (corresponding to all $n$ examples). The optimization problem is:

$$\min_{q \in \mathbb{R}^n, b \in \mathbb{R}^{2m(c+1)}} \quad \frac{1}{2}(q - \eta\mathbf{1})^T U (q - \eta\mathbf{1}) + V \cdot b$$

$$\text{s.t.} \quad Aq \leq b$$
$$\mathbf{1}^T q = \eta n$$
$$q \leq Q$$
$$q, b \geq 0,$$

where $U$ is a diagonal matrix containing the utilities of all examples and $A$ is a matrix formed from a constraints in (3) (a Numpy-based implementation of the columns of $A$ is shown in Figure 7 (right)). We denote the $i$-th column of $A$ by $a^{(i)}$ and refer to it as the "bias vector".

The Lagrangian is:

$$L(q, b, v, \mu, \alpha, \beta) = \frac{1}{2}(q - \eta\mathbf{1})^T U (q - \eta\mathbf{1}) + Vb + v^T(Aq - b) + \mu(\mathbf{1}^T q - \eta n) - \beta^T q - \delta^T b + \alpha^T(q - Q\mathbf{1})$$

subject to $v, \beta, \delta, \alpha \geq 0$.

Taking the gradient w.r.t. $q$ and setting it to zero:

$$q = \eta\mathbf{1} - U^{-1}\left(A^T v + \mu\mathbf{1} - \beta + \alpha\right). \tag{7}$$

Setting the gradient w.r.t. $b$ to zero gives us:

$$0 \leq v \leq V\mathbf{1}.$$

Plugging the expression for $q$ back into $L$, we have the minimization problem:

$$\min_{0 \leq v \leq V\mathbf{1}, \mu} \sum_{i=1} \inf_{\beta, \alpha \geq 0} \left\{ \frac{1}{2u_i}\left(v^T a^{(i)} + \mu + \alpha - \beta\right)^2 - \eta v^T a^{(i)} - \eta(\alpha - \beta) + Q\alpha \right\}. \tag{8}$$

We will use the following fact:

**Lemma 1.** $\omega^\star = [\lambda t + \xi]^+$ *is a minimizer to the objective function:*

$$\min_{\omega \geq 0} \left\{ \frac{1}{2\lambda}(\xi - \omega)^2 - t\omega \right\}.$$

Returning to (8), we have by complementary slackness (Boyd & Vandenberghe, 2004) that either $\alpha = 0$ or $\beta = 0$ so we consider each case separately.

**Case I:** If $\alpha = 0$, the minimizer $\beta^{(i)}$ to each of the inner optimization problems is by Lemma 1 $\beta^{(i)} = [w^{(i)} - \eta u^{(i)}]^+$, where $w^{(i)} = v^T a^{(i)} + \mu$. By plugging this into (8) and using (7), we have:

$$q^{(i)} = \eta - \frac{w^{(i)} - \beta^{(i)}}{u^{(i)}}, \tag{9}$$

and the stochastic gradients:

$$\nabla_v = -\frac{q^{(i)}}{\eta} a^{(i)}, \qquad \nabla_\mu = 1 - \frac{q^{(i)}}{\eta}. \tag{10}$$

**Case II:** If $\beta = 0$, the minimizer $\alpha^{(i)}$ to each of the inner optimization problems is by Lemma 1 $\alpha^{(i)} = [u^{(i)} (\eta - Q) - w^{(i)}]^+$. Note that we indeed have that $\alpha^{(i)} \cdot \beta^{(i)} = 0$ as expected. Similar to before, we have:

$$q^{(i)} = \eta - \frac{w^{(i)} + \alpha^{(i)}}{u^{(i)}}, \tag{11}$$

and the same stochastic gradients as in (10). From this, Algorithm 1 follows. By strong duality (Boyd & Vandenberghe, 2004), the solution returned by Algorithm 1 is the optimal solution to the minimization problem in (4).

### A.2.2 PROOF OF PROPOSITION 2

As shown in Appendix A.2.1, Algorithm 1 minimizes the loss in (8) via stochastic gradient descent. Writing the optimization variable as $w = (v, \mu)$, the stochastic gradient $g$ in (10) is bounded in norm by:

$$||g||_2 \leq \sqrt{\frac{Q^2}{\eta^2} ||\mathbf{a}||^2 + \left(1 - \frac{Q}{\eta}\right)^2} \leq \sqrt{\frac{Q^2}{\eta^2} (1 + ||\mathbf{a}||^2)} \tag{12}$$

because $0 \leq \mathbf{q} \leq Q$ in all iterations (see (5)), and the second inequality follows from the fact that $Q \geq \eta$ (maximum must be larger than the mean). The bias configuration $\mathbf{a}$ satisfies $||\mathbf{a}||_\infty = 1 + \epsilon$, where $\epsilon = \max\{\epsilon_D, \epsilon_R\}$. Hence: $||\mathbf{a}||^2 \leq (2(1 + \epsilon)m(c + 1))^2$ because $\mathbf{a} \in \mathbb{R}^{2m(c+1)}$. Since $\epsilon \leq 1$ (otherwise the constraints are void) while $m, c \geq 1$, we have the simplified bound:

$$||g||_2 \leq \frac{5Qm(c + 1)}{\eta}. \tag{13}$$

Denote the optimal solutions $v^\star$ and $\mu^\star$ and write: $R = ||v^\star||^2 + (\mu^\star)^2$. Using the learning rate schedule $\tau_t = 1/\sqrt{t}$ and the well-known convergence rates (Boyd & Mutapcic, 2008), we have:

$$\min_t \mathbb{E}[F_t] - F^\star \leq \frac{R^2 + \left(\frac{5Qm(c+1)}{\eta}\right)^2 \sum_t \tau_t^2}{2 \sum_t \tau_t} = O\left(\left(\frac{Qmc}{\eta}\right)^2 \frac{\log t}{\sqrt{t}}\right),$$

where $F_t$ is the loss function at iteration $t$.

### A.2.3 EMPIRICAL VERIFICATION

To verify the algorithm, we run it with the following constraints: (1) balance perceived gender attribute, (2) balance the age group attribute, (3) remove correlations between perceived gender and occupation, and (4) remove correlations between perceived gender and age. The goal is to satisfy all constraints simultaneously. In this illustrative example, however, we consider an attribute present if it is detected in any modality. Within each sensitive attribute axis, we set the target distribution to be the median distribution across all categories (i.e. 12% for both males and females and 1% for all age groups). Figure 8 displays a summary of the results. As shown in the figure, the algorithm mitigates both types of bias simultaneously.

### A.2.4 COMPARISON AGAINST PRE-PROCESSING ALGORITHMS

We compare, for reference, the data balancing algorithm against prior preprocessing in the classical binary classification setting.

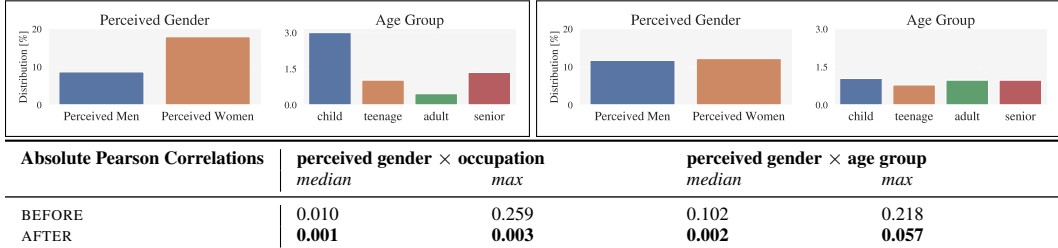

| Absolute Pearson Correlations | perceived gender $\times$ occupation | | perceived gender $\times$ age group | |
|---|---|---|---|---|
| | *median* | *max* | *median* | *max* |
| BEFORE | 0.010 | 0.259 | 0.102 | 0.218 |
| AFTER | **0.001** | **0.003** | **0.002** | **0.057** |

Figure 8: TOP: A comparison of the distribution of subgroups before and after applying the data diversity algorithm in Figure 7 in which we have four sets of constraints (see Section A.2.3). The target distribution is 12% for both perceived gender categories and 1% for all age groups. BOTTOM: Pearson correlation scores (max and median) between perceived gender and occupation/age. Algorithm 1 mitigates both first-order and second order biases simultaneously.

Table 3: Comparison of the data balancing algorithm against prior bias mitigation methods in the binary classification setting: correlation remover (CR), reduce-to-binary (R2B), reduction to cost-sensitive rules, threshold-optimizer (TO), randomized threshold optimizer (RTO). See Section A.2.4 for details. Last row is for the proposed data balancing algorithm.

| | | **Adult** | | | **Default on Credit Card** | | |
|---|---|---|---|---|---|---|---|
| | | *DP* | *Error* | *Balanced Error* | *DP* | *Error* | *Balanced Error* |
| | Baseline | $18.6 \pm .2$ | $14.5 \pm .1$ | $12.7 \pm .1$ | $02.1 \pm .3$ | $17.5 \pm .0$ | $17.6 \pm .0$ |
| *in-* | Reduction | $01.0 \pm .0$ | $16.4 \pm .1$ | $14.9 \pm .1$ | $01.4 \pm .1$ | $17.9 \pm .2$ | $18.2 \pm .1$ |
| *post-* | TO | $00.9 \pm .3$ | $19.9 \pm .0$ | $20.8 \pm .1$ | $00.7 \pm .3$ | $18.3 \pm .1$ | $18.8 \pm .1$ |
| *post-* | RTO | $00.8 \pm .2$ | $16.9 \pm .1$ | $16.5 \pm .0$ | $03.3 \pm .2$ | $20.7 \pm .1$ | $20.7 \pm .2$ |
| *pre-* | CR | $18.9 \pm .1$ | $14.7 \pm .2$ | $12.9 \pm .2$ | $00.8 \pm .2$ | $19.6 \pm .3$ | $20.0 \pm .2$ |
| *pre-* | R2B | $08.3 \pm .1$ | $15.4 \pm .0$ | $13.4 \pm .1$ | $00.9 \pm .1$ | $28.8 \pm .2$ | $29.1 \pm .2$ |
| *pre-* | **Balancing** | $09.1 \pm .2$ | $15.6 \pm .2$ | $13.7 \pm .2$ | $01.2 \pm .1$ | $17.9 \pm .3$ | $18.1 \pm .2$ |

Specifically, we evaluate the algorithm on the Adult income dataset (Kohavi, 1996) and the Default of Credit Card Clients (DCCC) dataset (Yeh & Lien, 2009), both from the UCI ML Repository (Blake & Merz, 1998) and are among the most widely used benchmark datasets in the ML fairness literature (Fabris et al., 2022). The Adult dataset contains 48,842 records with 14 attributes and the goal is to predict a binary summary of income level. The DCCC dataset contains 30,000 records with 24 attributes, where the goal is to predict if a client will default on credit card payment. In both datasets, binary sex is used as a sensitive attribute.

Besides preprocessing methods, we also include in-processing and post-processing in our comparison. However, the intent here is *not* to compete, since our goal is to balance the data prior to training.

We compare against the correlation remover method (Dudik et al., 2020) and reduce-to-binary (Alabdulmohsin et al., 2022a), reduction to cost-sensitive rules (Agarwal et al., 2018), threshold optimizer (Hardt et al., 2016), and randomized threshold optimizer (Alabdulmohsin & Lucic, 2021), and use the implementations released by authors or in the FairLearn package (Dudik et al., 2020). All hyperparameters are kept to their default setting. The base classifier is a two-layer MLP with 128 hidden units, ReLU activations (Nair & Hinton, 2010), that optimizes the log-loss using Adam (Kingma & Ba, 2014) with an initial learning rate of 0.001.

For evaluation metrics, we report demographic parity (DP), classification error rate, and the balanced error rate (average error rate across both subgroups).

As shown in Table 3, while data balancing as a mitigation option does not outperform in- and post-processing methods, the data balancing algorithm performs on par with other pre-processing techniques in mitigating bias without impacting model's quality. Importantly, our algorithm can handle a much broader setting than prior methods, including an arbitrary number of overlapping attributes and labels.

### A.3 MODEL CARD

Model details following Mitchell et al. (2019).

- **Model Architecture**: The model contains two towers, which are encoder-only vision transformers (ViT) (Dosovitskiy et al., 2020) for image and text modalities. We use architecture sizes ViT-S and ViT-B. For the image tower, we use either a patch size of $32 \times 32$ or $16 \times 16$. For the text tower, we use SentencePiece Kudo & Richardson (2018) tokenizer. The two towers are trained to maximize the similarity of image-text pairs via the contrastive loss.

- **Inputs**: The model takes both an image and a text as input.

- **Outputs**: The model outputs representations for image and text inputs.

- **Intended Use**: The primary use is research on multimodal applications, such as zero-shot classification and retrieval. We use such models to study the effectiveness of data balancing as a bias mitigation strategy.

- **Known Caveats**: As noted in several prior works, multimodal systems can pick up societal biases. While we demonstrate some of those issues in this work, our analysis is necessarily limited since fairness is a societal concept that cannot be reduced to simple statistical metrics.

- **System Description**: The model is analyzed in a stand-alone setting, and not used as part of a larger system.

- **Upstream Dependencies**: None.

- **Downstream Dependencies**: None.

- **Hardware & Software**: The model is implemented using JAX (Bradbury et al., 2018), Flax (Heek et al., 2020), and Big Vision (Beyer et al., 2022). It is trained on TPU v2.

- **Compute Requirements**: Each model is trained on $8 \times 8$ TPU v2 chips on 1B seen image-text pairs. A typical training run takes 1.3 days.

- **Model Initialization**: The model is trained from a random initialization.

- **Model Size**: ViT-S models have approximately 30M parameters per modality. ViT-B models have approximately 100M parameters per modality.

- **Training Dataset**: We use an English-only subset of WebLI (Chen et al., 2022), which comprises of images with alt-texts from the public web. See (Chen et al., 2022) for an overview of the data collection process.

- **Evaluation Datasets**: We evaluate the models on ImageNet-ILSRCV2012 (Deng et al., 2009), FairFace (Karkkainen & Joo, 2021), UTKFace (Zhang et al., 2017), and MIAP (Schumann et al., 2021).

## A.4 TRAINING CONFIGURATION

```
# input
batch_size = 16_384
shuffle_buffer_size = 250_000
pp = 'decode|resize(224)|value_range(-1,1)'

# model
model.temperature_init = 10.0

# optimizer
optax_name = 'scale_by_adafactor'
grad_clip_norm = 1.0
lr = 0.001
wd = 0.0001

# learning rate schedule
schedule.decay_type = 'rsqrt'
schedule.timescale = 5_000
schedule.warmup_steps = 5_000
```

## A.5 PROXIES

### A.5.1 LIST

We queried a large language model (LLM) to provide a suggested list of proxies that might link perceived gender with occupation. The LLM recommended the list shown in Table 4.

| | | | | | | |
|---|---|---|---|---|---|---|
| bench | briefcase | book | board | tools | desk | computer |
| suit | gun | handcuff | tray | ashtray | food | coat |
| goggles | whistle | clipboard | watch | flour | gym | sport |
| scrubs | microscope | credit card | shopping | sewing | machine | cabinet |
| calculator | paint | backpack | headset | microphone | camera | helmet |
| vacuum | glassware | car | comb | scissor | toothbrush | phone |

Table 4: Full list of proxies used in the experiments.

It justified each example by linking it to a particular occupation. For example, "bench" might provide a link to the occupation "judge," "briefcase" might provide a link to the occupation "manager," "book" might provide a link to the occupation "author," and so on.

### A.5.2 CAUSAL ANALYSIS OF DATA BALANCING APPROACHES

In this section, we analyze the properties of the `balanced` and `proxies` datasets from a causal perspective, using the formalism presented in Veitch et al. (2021), and make a more formal argument motivating the use of proxies. Specifically, we highlight a case where balancing labels $\mathbf{y}$ on sensitive attributes $\mathbf{s}$ alone is not enough because there are unbalanced proxies $\mathbf{w}$.

**Causal Model** We assume that the generative process for images is anti-causal with respect to the label $\mathbf{y}$, sensitive attribute $\mathbf{s}$, and proxy $\mathbf{w}$ variables in our setting. In other words, we assume that these variables are underlying causes of the image/text pair $\mathbf{x}$ that is taken as input.

In this setting, Veitch et al. (2021) show that the input $\mathbf{x}$ can then be decomposed into three pieces, illustrated in Figure 9(a):

- $\mathbf{x}_{\mathbf{s}\perp}$, or parts of $\mathbf{x}$ that are not causal descendants of the senstiive attributes $\mathbf{s}$. Importantly, $\mathbf{x}_{\mathbf{s}\perp}$ includes portions of $\mathbf{x}$ that are causal descendants of $\mathbf{y}$, but not of $\mathbf{s}$.
- $\mathbf{x}_{\mathbf{y}\perp}$, or parts of $\mathbf{x}$ that are not causal descendants of $\mathbf{y}$.
- $\mathbf{x}_{\mathbf{y}\wedge\mathbf{s}}$, or parts of $\mathbf{x}$ that are causal descendents of both $\mathbf{y}$ and $\mathbf{s}$

The features $\mathbf{x}_{\mathbf{s}\perp}$ are often considered "core" features, since they are not causally influenced by the sensitive attributes $\mathbf{s}$. Veitch et al. (2021) show that predictors that are only a function of $\mathbf{x}_{\mathbf{s}\perp}$ can exhibit desirable counterfactual invariance properties. Correspondingly, the features $\mathbf{x}_{\mathbf{y}\perp}$ are often considered "spurious" features, since they have no causal relationship to the labels $\mathbf{y}$. Finally, the

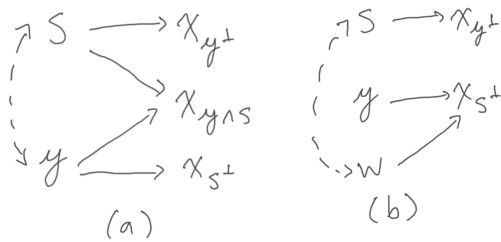

Figure 9: DAGs describing causal models for image/text pairs **x** as a function of sensitive attributes **s**, labels **y**, and proxies **w**. Solid arrows represent causal relationships; bidirectional dashed arrows represent non-causal relationships introduced by confounding or selection. (a) is a general generative model without proxies; (b) is a simplified generative model with proxies.

intersection information $\mathbf{x}_{\mathbf{y} \wedge \mathbf{s}}$ entangles information about both **y** and **s**; as we show below, this can introduce complications for data balancing mitigations.

**Cases Where Balancing Proxies is Needed**    We now provide a formal example highlighting the potential role of proxies in the data balancing algorithm introduced in the paper. For simplicity, in this argument we will assume that there is no intersection information $\mathbf{x}_{\mathbf{y} \wedge \mathbf{s}}$, because it is sufficient to show how proxy variables can complicate data balancing. Veitch et al. (2021) call this the "purely spurious" case.

As shown in Makar et al. (2022), in this setting, data balancing can be sufficient to achieve desirable predictive invariances, *if there are no uncontrolled confounders*. However, if there are proxy varaibles **w** that are (1) correlated with the sensitive attributes **s**, and (2) correlated (causally or not) with the "core" features $\mathbf{x}_{\mathbf{s} \perp}$, these can operate as confounders that introduce correlations between sensitive attributes **s** and optimal predictors $f(\mathbf{x})$ designed to predict labels **y** from inputs **x**. Importantly, this can happen even when labels **y** and sensitive attributes **s** are marginally independent, as we might achieve with data balancing.

To see how this can occur, consider the DAG in Figure 9(b) where we introduce proxies **w** with a bidirectional arrow connecting them to **s**. Here, we assume that there is no bidirectional edge connecting **y** to **s**, or connecting **y** to **w**, due to data balancing. [2] Even though these edges are absent, the **s**–**w** edge creates an open path between **s** and $X_{\mathbf{s} \perp}$. This path implies that $\mathbf{x}_{\mathbf{s} \perp}$, the features that are causally influenced by **y**, but not by **s**, can nonetheless be correlated with **s**. Any reasonable $f(\mathbf{x})$ will incorporate information from the core features $\mathbf{x}_{\mathbf{s} \perp}$, so this correlation implies that $f(\mathbf{x})$ will have some dependence on **s**. Thus, conditioning on **s** can change the distribution of $\mathbf{x}_{\mathbf{s} \perp}$, resulting in a different conditional expectation $E[f(\mathbf{x}) \mid \mathbf{s}]$ across levels of **s**.

Note that this dependence path is cut off if the edge between **s** and **w** is eliminated, e.g., by balancing proxies **w** with respect to sensitive attributes **s**, as done by `proxies`. This is the primary motivation for this approach at a population level. However, there are a number of potential complications to this argument. Chief among these is the need to balance on an exhaustive set of proxy variables to eliminate the backdoor path described above. Notably, there is no guarantee that balancing an incomplete set of proxies will reduce dependence; indeed, in some cases the backdoor dependence can increase, a phenomenon known as "bias amplification" in the causal inference literature (see, e.g., Middleton et al., 2016). In addition, this argument is made a the population level, but in practice, finite sample complications in optimization or sampling variability may dominate the benefits of proxy balancing in terms of association bias.

---

[2] In practice, after data balancing, there can be an edge connecting **y** to **w**, which would imply that there is an open path between **y** and **s**, even though there is no correlation between **y** and **s** because these correlations are made to cancel by the data balancing algorithm. This complication is out of scope for this motivating example.

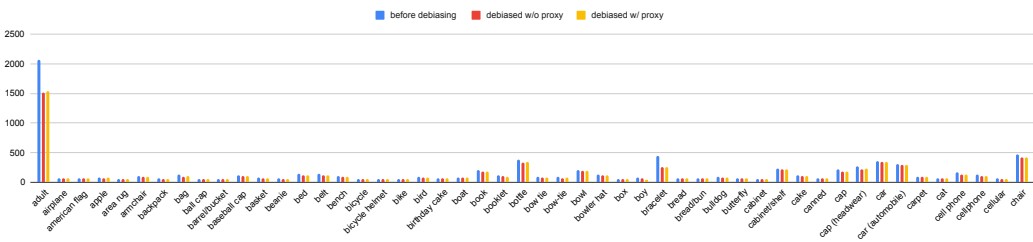

Figure 10: The distribution changes of top-50 pseudo object labels before and after debiasing.

Table 5: Example COCO captions with human mentions.

| Keyword | Caption with human mentions |
|---------|------------------------------|
| adult | Group of adults at outdoor gathering on clear day. |
| boy | A little boy sitting on a wooden bench eating half a sandwich. |
| bride | A bride is standing on the grass under an umbrella. |
| child | A woman playing catch with her young child. |
| children | Children are playing soccer on a field with several adults observing nearby. |
| daughter | The mom is teaching her daughter to play baseball |
| elderly | A woman and an elderly woman sitting at a desk together in a classroom setting with notebooks and pencils on the desk. |
| father | Baby girl feeding her father some pink and white birthday cake. |
| female | a female in a white shirt is playing tennis |
| gentleman | Two gentleman in formal suits, one of them is adjusting the collar of the other. |
| girl | two girls in red shirts grass and a baseball glove |
| groom | The cat grooms itself atop the workshop table. |
| lady | a lady sitting in a half shell hut |
| male | A male cook standing by the kitchen sink. |
| man | a man with a bat looks into the air at a pop up |
| men | a couple of men that are walking around on some grass |
| mom | A dog and a cat share a moment |
| mother | A mother elephant and her calf touching trunks in the tall grass |
| people | some people on some grass playing frisbee and trees |
| player | A black and white photo of baseball players looking for a ball |
| sister | Two sisters watch as their kids decorate a cake |
| son | Mom gives her daughter a lesson in using her baseball glove. |
| teen | Three teenagers are throwing a frisbee back and forth in a field. |
| woman | A woman on a tennis court serves the ball. |

## A.6 MODEL QUALITY ANALYSIS

To understand the discrepancy of model quality on classification and retrieval tasks, we conduct a in-depth study on the data distribution before and after debiasing. More specifically, we analyze the distributions on image modality (pseudo object labels, OCRs...), text modality (text length, word frequency, Part-Of-Speech tags...) and image-text cross-modality (image-text similarity). Surprisingly, debiasing doesn't change most distributions, except the pseudo object labels (Figure 10), which are annotated by the open-vocabulary detector OWL-ViT (Minderer et al., 2022b) and reduce around 25% "adult" after the debiasing.

We further analyze the test sets of ImageNet-ILSRCV2012 (Deng et al., 2009) and COCO Captions (Chen et al., 2015), and find that the majority of ImageNet images are not associated with human-related labels. On the contrary, about 40% COCO images have captions mentioning humans (Table 5).

To verify the hypothesis that the model quality change is primarily caused by the human distribution shift in the training data, we randomly drop examples (instead of relying on the debiasing algorithm) in which humans occur in either image or text. In Figure 11, dropping more human examples leads to higher classification but lower image-text retrieval numbers, which perfectly reproduces the debiasing effect on the model quality.

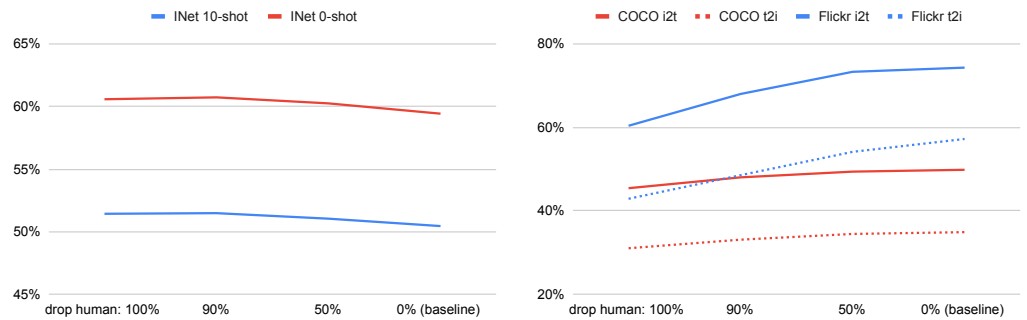

Figure 11: The percentage of human training examples (mentioning human in image or text) affects the model performance on both classification (INet 10-shot and 0-shot) and retrieval (COCO and Flickr 0-shot) tasks. All numbers are the mean values of 3 runs.

Furthermore, we neutralize the human mentions in text (e.g. by replacing "female"/ "gentlemen" / "she" with "person" / "people" / "the person"), in order to preserve training examples (i.e. avoid dropping human examples) and mitigate the side-effect of debiasing on model quality. Figure 12 demonstrates the feasibility of such an approach, and we would like to leave this as a future work.

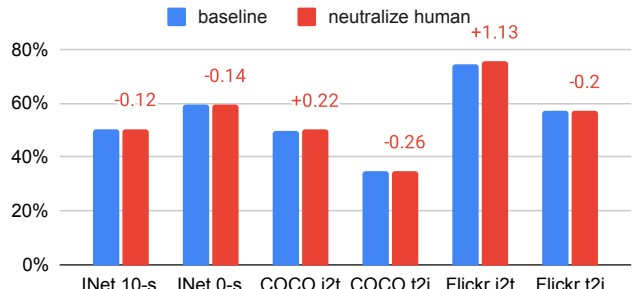

Figure 12: Human neutralization minimizes the influence to model quality.

## A.7 ADDITIONAL FIGURES

### A.7.1 REPRESENTATION BIAS

Figures 13 and 14 report the median scores (as opposed to the means) for the analysis of Section 4.1.

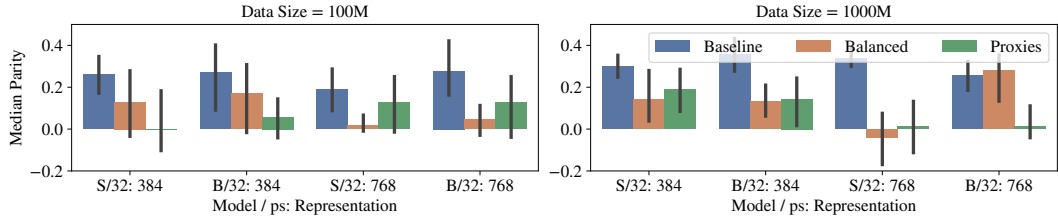

Figure 13: Median parity $\mathbb{E}[p(\text{man}) - p(\text{woman})]$ across images from the ILSRCV2012 dataset (Deng et al., 2009). See Figure 2. Values closer to zero are better.

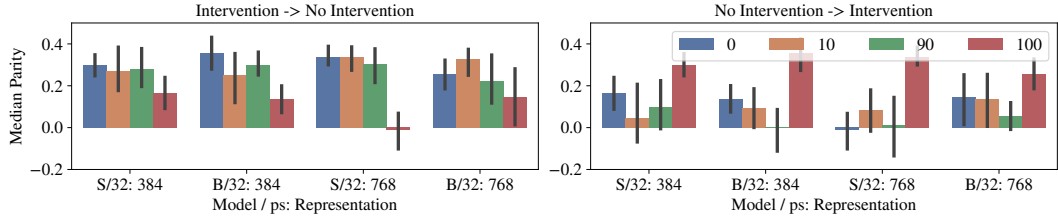

Figure 14: Similar to Figure 3 but using the median scores, instead of the mean.

### A.7.2 ASSOCIATION BIAS

Figures 15 and 16 report the median association bias scores (as opposed to the means) for the analysis of Section 4.2.

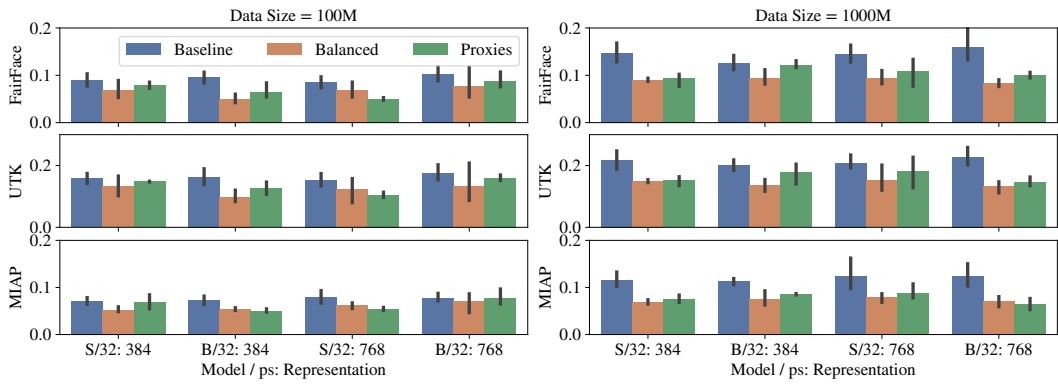

Figure 15: Reporting the median bias scores instead of the means for association bias. See Figure 4 and Section 4.2.

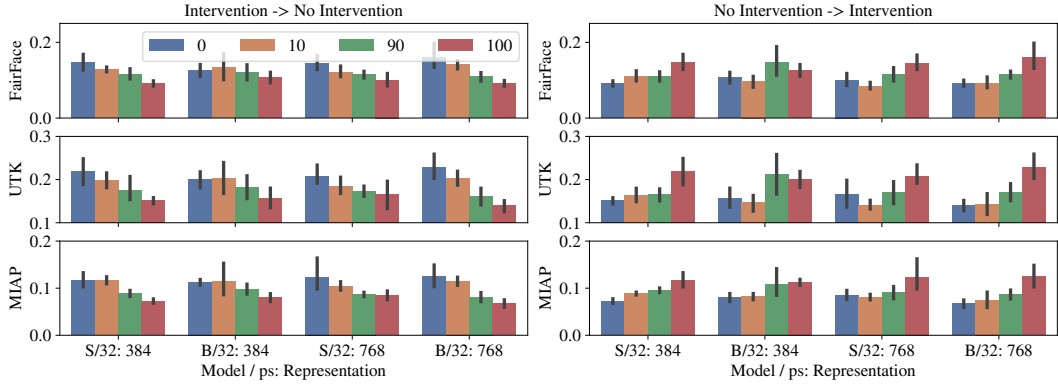

Figure 16: A summary of how CLIP learns or unlearns association bias (shown in $y$-axis) when intervened data comprises different percentages [%] of training duration. Here, we report the median bias scores across occupations instead of the mean, which was reported in Figure 5

### A.7.3 MODEL QUALITY

Figures 17, 18, and 19 report the full model quality evaluation results in terms of few-shot classification, zero-shot classification, and retrieval.

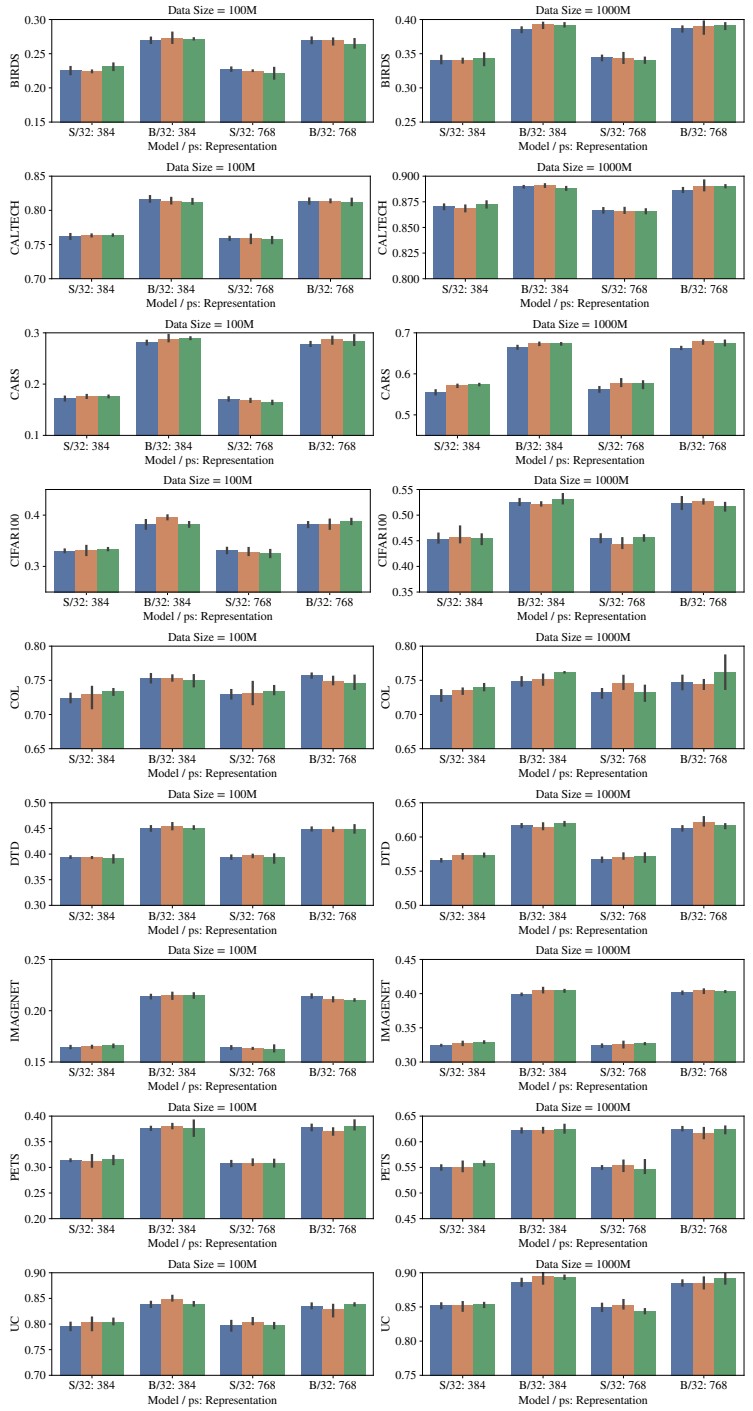

Figure 17: Full 10-shot evaluation results across the nine datasets in Table 1.

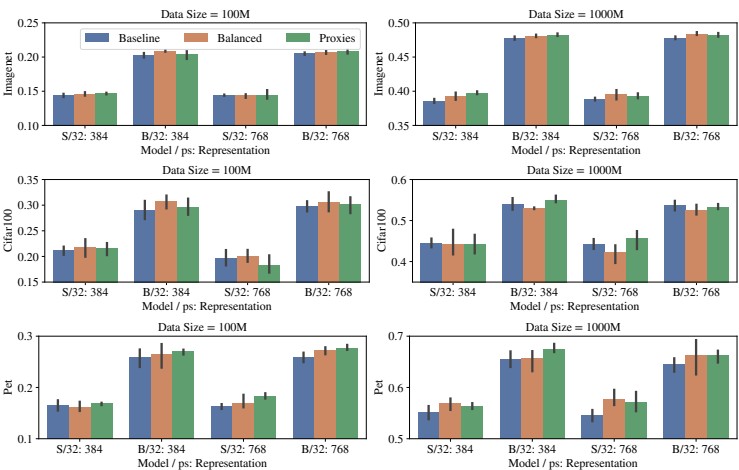

Figure 18: Full zero-shot evaluation results across the three datasets in Figure 6.

### A.7.4 ENCODING SENSITIVE ATTRIBUTES

Figure 20 reports the full evaluation results of predicting sensitive attributes. See Section 4.3 for details.

Table 6: Association bias (AB) between gender and occupation in SigLIP models when they are pretrained with or without data balancing on a high-quality subset of WebLI (Chen et al., 2022) (see Appendix A.8).

| Data | FairFace | UTK | MIAP |
|------|----------|-----|------|
| Without Intervention | 38.8 | 38.2 | 28.4 |
| With Intervention | **29.9** | **33.7** | **20.5** |

Table 7: Classification and retrieval evaluation results [%] in SigLIP ViT-B/16 with and without data balancing. Data balancing seems to *improve* the model quality in this setting.

| Data | ImageNet | | COCO Retrieval | | Flickr Retrieval | |
|------|----------|--------|----------------|----------------|------------------|----------------|
| | **10-shot** | **0-shot** | **img2txt@5** | **txt2img@5** | **img2txt@5** | **txt2img@5** |
| Without Intervention | 70.7 | 77.0 | 86.0 | 73.4 | 98.5 | 94.3 |
| With Intervention | **71.4** | **77.5** | **87.1** | **73.7** | **98.8** | **94.9** |

## A.8 IMPACT OF DATA QUALITY AND ARCHITECTURAL IMPROVEMENTS

In this section, we investigate the impact of data quality and architectural improvements on model performance. Rather than training CLIP on minimally filtered image-text pairs (where filtering primarily addresses personally identifiable information (PII) and inappropriate content as described in (Chen et al., 2022), we introduce an additional quality filtering step. This step leverages pretrained models to assess the semantic alignment between image and text content. Image-text pairs with low alignment scores are discarded.

In this evaluation, we use the recently-introduced Sigmoid-loss Language Image Pretraining (SigLIP) (Zhai et al., 2023). We train a two-tower SigLIP with ViT-B/16 image encoder and ViT-B text encoder for 10B examples with and without data balancing (with proxies). Because SigLIP does not provide calibrated probability scores, we evaluate association bias (AB) in the model between gender and occupation by drawing one image of a man and one image of a woman unifromly at random, and asking the model to predict which of the two images contains a person with a given occupation (e.g. "nurse"). Parity is then defined to be the difference in probability that the model chooses one particular gender over the other: $|p(\mathrm{man}) - p(\mathrm{woman})|$.

Table 6 provides the maximum observed parity in each model across all occupations. Clearly, data balancing has a positive impact on the model's bias, in agreement with our earlier results in Section 4. Interestingly, however, we do not observe a negative impact on the model quality this time. In fact, the quality of the model seems to improve in all metrics, as shown in Table 7. This suggests that data quality and architectural improvements can mitigate the potential negative impact of data balancing, allowing us to reduce biases in the model without impacting its quality.

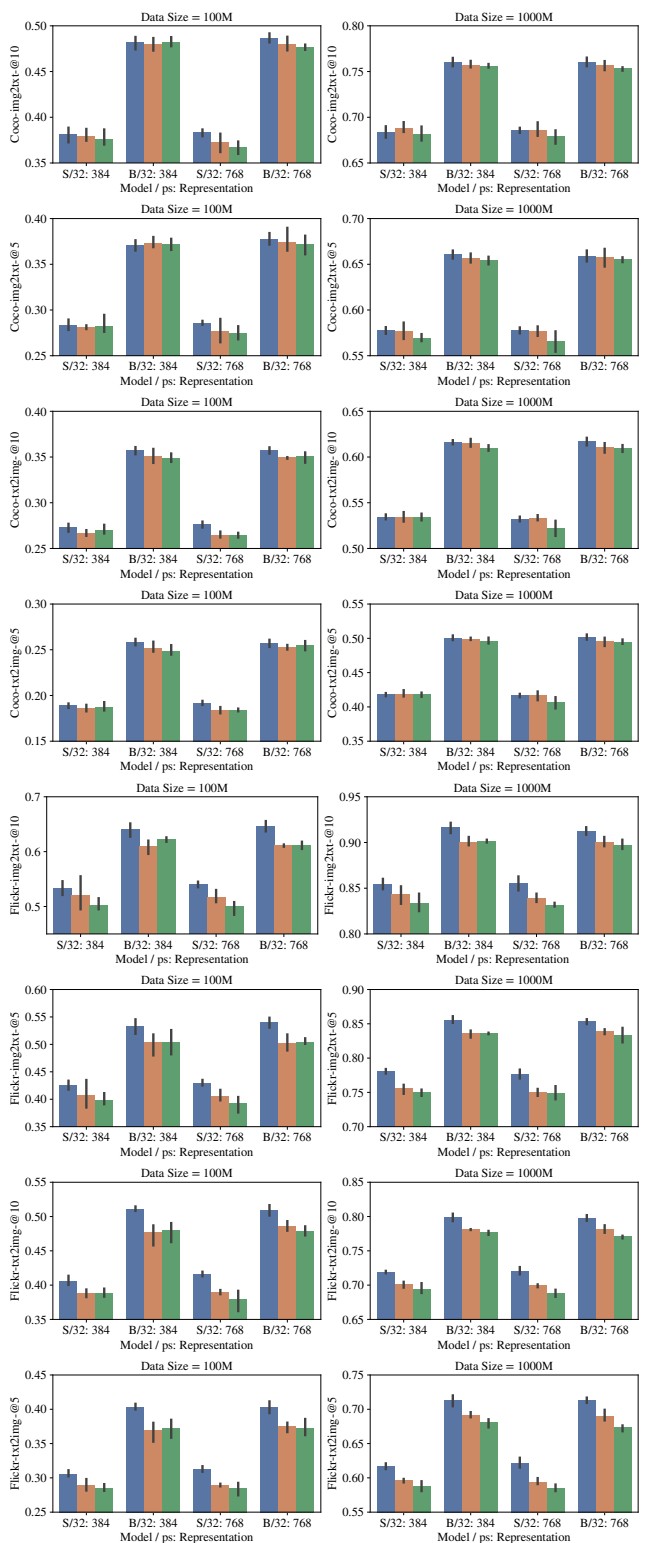

Figure 19: Full retrieval results across the two datasets in Figure 6.

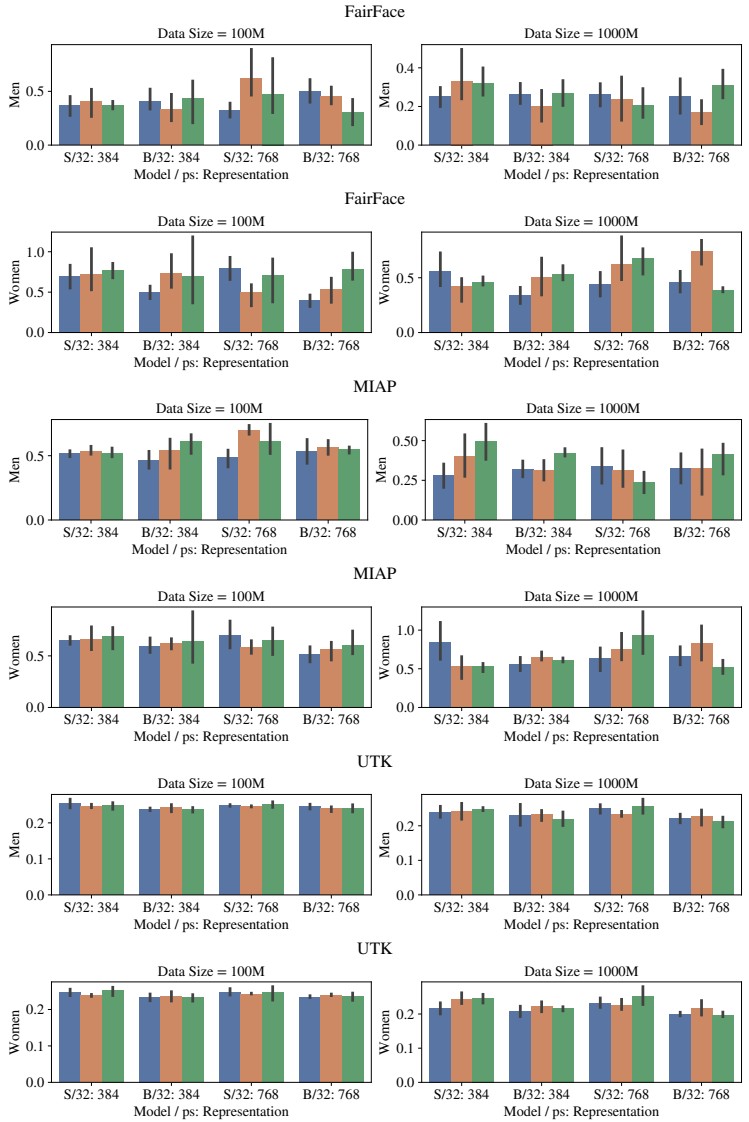

Figure 20: Full evaluations results for predicting sensitive attributes in Fairface, UTKFace, and MIAP, disaggregated by perceived gender. The $y$-axis the mean log-loss (lower is better). See Section 4.3 for details.

