# OpenReview forum: "CLIP the Bias: How Useful is Balancing Data in Multimodal Learning?"
_ICLR.cc/2024/Conference — ICLR 2024 poster_

### Official Review · Reviewer_yvnK · 2023-10-28

**Soundness:** 3 good
**Presentation:** 3 good
**Contribution:** 2 fair
**Rating:** 6
**Confidence:** 4

**Summary:**

This study delves into the efficacy of data-balancing in reducing biases in CLIP models, highlighting its strengths and constraints. A novel data-balancing algorithm is introduced, showing success in mitigating representation biases, particularly through fine-tuning, but offering lesser impact on association biases, with a variable effect on task performance such as enhanced classification yet impaired retrieval. The research culminates with strategic guidance aimed at optimizing data balancing in multimodal learning systems.

**Strengths:**

1. Nice definition of various types of bias in data or model, this is a solid foundation to understand the problem

2. Given a nice definition of bias such as gender, this work shows the effect of model training with various type settings. (More or less data, fine tuning with various length of training time)

3. Tested on various datasets and backbone designs.

4. The proposed balancing algorithm does seem to successfully diminish bias without compromising the quality of the model.

**Weaknesses:**

1. I hope to see figures that are easier to interpret.

- For Figure 2 (top): It seems you intend to demonstrate that even with extra training data, bias persists. I struggled to determine which bars were being compared. A similar issue occurs with Figure 3.
- Regarding Figure 4: At first glance, without referring to the captions, all the color bars appear identical. My initial interpretation was that the results were largely uniform across the board.


2. I find myself somewhat perplexed by the conclusions drawn from the study's results. This confusion does not necessarily point to a flaw in the research but suggests that further clarification might be beneficial. I recommend referring to the detailed queries I've raised in the question section.

**Questions:**

1. In Section 4.1, the authors discuss various strategies, including adjusting the training set sizes and fine-tuning with or without dataset intervention.

- While these strategies are standard in model training, the authors' approach underscores a well-known principle: merely increasing the size of a 'corrupted' dataset—say, by tenfold—will not address inherent issues. This scenario is a classic case of "garbage in, garbage out."

- Furthermore, the practice of fine-tuning a model on a specific dataset inevitably alters some pre-existing weights, adapting the model to new data characteristics. Consequently, it is not surprising that models fine-tuned on intervened sets exhibit improved bias metrics.

- In summary of this question: What's new here?

2. This study suggests that balancing our training datasets can mitigate bias. However, it also implies that researchers must painstakingly identify what constitutes bias and determine the features requiring adjustment. I am particularly interested in the authors' insights on how their proposed framework accommodates additional features present in real-world datasets.


3. My concern extends to the interrelated nature of certain features, as highlighted in the occupation versus gender discussion in Section 4.2. While it is inappropriate to rely on stereotypes for inferring gender, assuming a 50/50 gender split across all professions disregards real-world disparities. It remains unclear how the proposed methods address "association bias."
- A link to “U.S. BUREAU OF LABOR STATISTICS” https://www.bls.gov/cps/cpsaat11.htm

---

> ### Author Response · Authors · 2023-11-13
> **Response**
>
> Thank you for your thoughtful review of our paper. We greatly appreciate your positive remarks on the definitions of various types of bias and our approach of testing across different datasets and models. We acknowledge your constructive criticism regarding the clarity of our figures and the need for more explicit interpretation of our results. Please see our detailed response below.
>
> Addressing the concerns about the interpretability of our figures, in Figures 2 and 3, the bars that we compare are for "baseline", "balanced", and "proxies" within each experimental configuration. For example, we compare them when using 1B examples on ViT-B backbone with 768 representation length. But, as you correctly mention, we are also interested in comparing across other dimensions, such as the size of the training data to show that even with extra training data, bias persists. We will clarify this in the revised version of the paper.
>
> In Figure 4, both “balanced” and “proxies” have lower AB than the baseline in most of the experiments, with $p$-values of less than $10^{-6}$. The error bars are for the three random seeds used for each experiment, while the $p$-values are calculated across all experiments (e.g. seeds, model size, representation length).
>
> **Answers to Questions**
> * We believe our findings are more nuanced. For example, it might turn out that training on large datasets helps CLIP learn a more accurate representation of sensitive attributes and labels, thereby separating them. In our experiments, however, we answer that question by showing that a bigger dataset doesn’t fix the bias. Also, it was not clear how effective fine-tuning would be for AB and RB. Our experiments provide a detailed answer: RB is sensitive to the *last* distribution it sees on the data so fine-tuning is as effective as training on balanced data from the start, but in AB the effectiveness of fine-tuning is more gradual and depends on the length of its duration. We also study the impact of using proxies as well; e.g. proxies help RB but hurt AB, and we offer an explanation. We also study in detail the impact on the model's quality, and provide a detailed explanation in Appendix A.6 for why classification and retrieval metrics are impacted differently. We hope this clarifies our contribution.
>
> * We agree with your assessment. We hope that since our proposed algorithm is designed to be adaptable to various sensitive attributes and labels by relaxing many assumptions (e.g. attributes and labels can overlap), it is flexible enough to incorporate different scenarios that may be pertinent in real-world applications.
>
> * We acknowledge the complexity of this issue. As you mentioned, real-world disparities can continue to have an influence on the model even after balancing the data, and we observe evidence for this in our experiments. For example, AB is not removed entirely even after decorrelating perceived gender with occupation. But, we believe that understanding the extent of how effective balancing the data is for multimodal learning, such as CLIP, would be a step in the right direction.
>
> We hope these responses address your concerns and clarify our approach. Again, we are grateful for your constructive feedback, which has highlighted areas for further exploration and improvement in our research.
>
> If there are any other questions or concerns, we would appreciate it if you let us know so we can respond to them during the discussion and improve the manuscript.
>
> Thank you

---

> ### Author Response · Authors · 2023-11-17
> **Follow-up**
>
> Dear reviewer,
>
> We would like to kindly inquire if our responses have satisfactorily addressed your concerns and questions. We are open to further clarification if needed. If we have addressed your concerns, we would appreciate it if you consider revising your review/score accordingly. Otherwise, please let us know so we can respond to your inquiries during the discussion period.
>
> Sincerely

---

### Official Review · Reviewer_5bEw · 2023-11-01

**Soundness:** 2 fair
**Presentation:** 3 good
**Contribution:** 2 fair
**Rating:** 6
**Confidence:** 3

**Summary:**

This paper studies two types of biases, representation biases (RB) and association biases (AB), in vision-language models such as CLIP.

RB refers to that the model learns to prefer sensitive attributes (e.g., gender, age groups) in the training data. AB refers to that the model associates certain concepts with sensitive attributes. (e.g. occupations with a specific gender)

Studying such biases is an important problem in the real-world as CLIP-like models are widely used in the industrial applications. In this paper, the authors first investigate the empirical evidence of both biases and how it transfers from data to the model. They show that RB is sensitive to the latest training data distribution thus fine-tuning (FT) is an effective approach to reduce RB. However, FT is weak on AB. They then propose a data balancing algorithm to alleviate both RB and AB by estimating an optimal weight for each data example.

**Strengths:**

1. The empirical evidence of RB and AB is well supported in the experiments.
2. The proposed data balancing algorithm is principled with theoretical analysis.
3. The paper studies an interesting and important problem which may have a wide impact in real-world industrial applications, such as recommender systems and advertising.

**Weaknesses:**

1. The number of sensitive attributes in the experiments is limited to only gender and occupation.
2. Further experiments on proposed data balancing algorithm is lacking in the main text.

**Questions:**

1. How exactly the de-correlation between sensitive attributes and proxies is implemented?
2. In AB experiment, why adding proxies has inconsistent performance?
3. In data balancing algorithm, how to intuitively interpret $\alpha$ and $\beta$? and how $s$ is determined, is it calculating from the dataset over sensitive attributes?

---

> ### Author Response · Authors · 2023-11-13
> **Response**
>
> Thank you for your constructive feedback. We appreciate your recognition of the empirical evidence provided, the principled approach of our data balancing algorithm, and the relevance of our study to real-world applications.
>
> Addressing your concerns regarding the limited number of sensitive attributes in our experiments, we acknowledge that our current focus has been primarily on perceived gender and occupation. This decision was intentional. First, we did not have skin tone annotations in the downstream evaluation datasets. For example, FairFace and UTKFace use race/ethnicity, not skin tone. Second, our goal was to determine how biases in the data (both RB and AB) transfer to their corresponding biases in the model. We believe that using perceived gender and occupation would still provide useful insights that can illuminate this relationship.
>
> About the experiments of the data balancing algorithm, those comparisons with classical preprocessing methods are available in Appendix A.2.  We prefer to keep them in the appendix since the primary scope of our work is to study the impact of balancing the data on CLIP models.
>
> **Answers to the Questions**
> * We treat proxies as labels, so they are included in the set of variables $\textbf{y}_r$ in Equation 3. The algorithm identifies the set of weights that satisfy all constraints, including the correlation constraints between the sensitive attributes and proxies. For instance, we remove any correlation between the proxy “computer” and perceived gender.
>
> * This phenomenon might be attributed to the complex interactions between different variables in the dataset. In [1], for example, the authors find that many factors influence the model’s prediction of sensitive attributes, such as gender, including non-obvious ones such as beliefs and communion. We will elaborate on this in the revised version of the paper. This is why we do statistical analysis to confirm the significance of our findings.
>
> * $\alpha$ and $\beta$ are Lagrange multipliers associated with the constraints $q\le Q$ and $q\ge 0$ respectively (in our experiments $Q=1$). One way to interpret Lagrange multipliers is using the “sensitivity analysis” method [2]. In brief terms, a large value of either $\alpha$ or $\beta$ means that the training example has a bigger “influence” on the final solution (set of weights). More precisely, a large value of $\alpha$ means that the training example will have a bigger impact on the optimal objective function if we allow weights to be larger than $Q$. We will elaborate on this in the revised version of the paper. On the other hand, $s$ is a binary indicator for the sensitive attribute. For example, if we have two attributes for perceived gender, then $s=(1, 0)$ for perceived women and $s=(0, 1)$ for perceived men.
>
> Again, we are grateful for your constructive feedback, which has highlighted areas for further exploration and improvement in our research.
>
> If there are any other questions or concerns, we would appreciate it if you let us know so we can respond to them during the discussion and improve the manuscript.
>
> Thank you
>
> [1] Fraser, Kathleen C., Svetlana Kiritchenko, and Isar Nejadgholi. "A Friendly Face: Do Text-to-Image Systems Rely on Stereotypes when the Input is Under-Specified?." arXiv preprint arXiv:2302.07159 (2023).
>
> [2] Stephen Boyd and Lieven Vandenberghe. Convex optimization. Cambridge University Press, 2004.

---

> > ### Comment · Reviewer_5bEw · 2023-11-20
> >
> > Thank you for taking the time address my concerns and answer my questions in detail. I understand that the absence of relevant, public datasets to conduct larger-scale experiments with more sensitive attributes.
> >
> > Regarding the current presentation of data balancing algorithms, from the reader's perspective, I think it could be great to add such intuitive explanations as you replied above. Because it's a bit easy to lose track for the first read for ones who don't have strong relevant background.
> >
> > Overall, I am satisfied with the clarifications, thus I'm raising my score given the concerning points are well-addressed and the questions are properly answered. Given the importance and wide impact of the problem studied in this paper, I would highly encourage you to continue refining the empirical evidence of such data-model bias transfers.

---

> ### Author Response · Authors · 2023-11-17
> **Follow-up**
>
> Dear reviewer,
>
> We would like to kindly inquire if our responses have satisfactorily addressed your concerns and questions. We are open to further clarification if needed. If we have addressed your concerns, we would appreciate it if you consider revising your review/score accordingly. Otherwise, please let us know so we can respond to your inquiries during the discussion period.
>
> Sincerely

---

### Official Review · Reviewer_yZxD · 2023-11-03

**Soundness:** 3 good
**Presentation:** 3 good
**Contribution:** 3 good
**Rating:** 6
**Confidence:** 4

**Summary:**

This paper identifies the societal bias issue in CLIP models and provides possible explanations and workarounds. Specifically, the representation and association biases are taken into consideration. While these biases can be somehow addressed with the data balancing strategy, other issue emerges. In this regard, this paper discusses in detail the role of data balancing in handling bias issues, and provides useful insights.

**Strengths:**

- The paper is well motivated and clearly written.
- Simple data balancing strategies are proposed to tackle the bias issue, demonstrating promising results.
- Comprehensive experimental results and analysis are presented, which may benefit the reader in relevant fields.

**Weaknesses:**

- While AB is relatively easy to mitigate, RB seems much more difficult to remove. In this regard, I would suggest the authors to shed more light on possible reasons and solutions. For example, I assume data augmentation shall be a promising workaround, and encourage the authors to explore more.

**Questions:**

- To me, the representation and association biases respectively correspond to the distribution shift of marginal output probability p(y) and conditional output probability p(y|x). What about another widely-discussed distribution shift in domain adaptation/generalization literature, i.e., marginal input probability p(x)? Can it also be a significant bias issue in large multimodal models?

---

> ### Author Response · Authors · 2023-11-13
> **Response**
>
> Thank you for your thoughtful evaluation of our paper and for your insightful questions and suggestions. We appreciate your recognition of the strengths of the paper, including the motivation, clarity of presentation, and the comprehensive experimental results we provided. Your feedback is invaluable in enhancing our work.
>
> Addressing your question regarding the mitigation of representation bias (RB) versus association bias (AB), we suspect that the reason AB is easier to mitigate is because of its closed-vocabulary nature. RB, on the other hand, deals with an open-vocabulary problem so it is harder to mitigate. Nevertheless, we still observe a positive message. As shown in Figure 2: (1) balancing the sensitive attributes generally helps, and (2) using proxies make balancing more effective. However, these do not eliminate RB entirely as acknowledged in the paper. We will elaborate on this more in the revised version.
>
> Your suggestion of developing a suitable augmentation strategy that might help RB is well-taken. Another open research question is to identify (from the statistics of the data) the minimal set of proxies that can have the biggest impact on RB. We plan to explore these directions in future work.
>
> About viewing AB and RB as distribution shifts, this is an interesting angle to view both problems. In your notation, $y$ would be the sensitive attribute and $x$ would be the label (e.g. occupation or proxy) if we understand it correctly. So, a shift in $p(y)$ would correspond to RB (balancing → uniform distribution) white a shift in $p(y|x)$ corresponds to AB (balancing → $p(y|x)=p(y)$, i.e. independence). It’s unclear, however, if a shift in $p(x)$ (i.e. changing the distribution of labels) would correspond to a notion of fairness/bias. We would appreciate it if you could elaborate on this.
>
> Again, we are grateful for your constructive feedback, which has highlighted areas for further exploration and improvement in our research.
>
> If there are any other questions or concerns, we would appreciate it if you let us know so we can respond to them during the discussion and improve the manuscript.
>
> Thank you

---

> ### Author Response · Authors · 2023-11-17
> **Follow-up**
>
> Dear reviewer,
>
> We would like to kindly inquire if our responses have satisfactorily addressed your concerns and questions. We are open to further clarification if needed. If we have addressed your concerns, we would appreciate it if you consider revising your review/score accordingly. Otherwise, please let us know so we can respond to your inquiries during the discussion period.
>
> Sincerely

---

> > ### Comment · Reviewer_yZxD · 2023-11-22
> >
> > Thanks for the detailed reply, which has mainly addressed my concerns.
> >
> > Regarding the perspective of distribution shifts, since multimodal learning is considered, can $p(x)$ also be interpreted as the visual input distribution? In this context, I would like your opinions on the possible connections to domain adaptation/generalization works that mainly aim at the bias in $p(x)$ between training and deployment.

---

### Meta-Review · Area_Chair_eYBT · 2023-12-13

**Metareview:**

This paper received unanimous slightly positive scoring, 6, 6, and 6. These ratings have been maintained after authors rebuttal.

Although this paper does not appear to present an original approach, it analyzes the bias issues from two perspectives in vision/language (CLIP) models, and provide insights on how to manage (rebalance) the data to achieve a better training. This study is corroborated by a thorough experimental analysis. In the end, this work is interesting to show to the community.

This paper is deemed acceptable for publication to ICLR 2024.

**Justification For Why Not Higher Score:**

Ratings are just above threshold; the paper does not seem to deserve more than a poster presentation.

**Justification For Why Not Lower Score:**

Unanimous positive evaluations, just above threshold. Unless the policy for ICLR 2024 is to raise the bar over rating 6, there is no reason to reject the paper.

---

### Decision · Program_Chairs · 2024-01-16

Accept (poster)